# SUIT: KNOWLEDGE EDITING WITH SUBSPACE-AWARE KEY-VALUE MAPPINGS

**Haewon Park**[*], **Sangwoo Kim**[*†], **Yohan Jo**[‡]
Graduate School of Data Science, Seoul National University
{dellaanima2,hemy0101,yohan.jo}@snu.ac.kr

## ABSTRACT

Knowledge editing aims to efficiently correct factual errors in language models. Widely used locate-then-edit methods update an MLP layer by adjusting its weights to change the mapping between the layer's input vector (key) and output vector (value), thereby editing the model's knowledge. As this update is driven by key and value vectors, obtaining these vectors without careful constraints causes significant model perturbations beyond the targeted edit, a common issue in many prior knowledge editing methods. To address this, we propose Subspace Knowledge Edit (SUIT), which computes key and value vectors only within the subspace of critical features relevant to the edit. Our empirical results on LLaMA3, GPT-J, and Qwen2.5 models show that SUIT dramatically improves knowledge preservation over strong baselines while maintaining high editing performance. These results support the claim that SUIT successfully identifies the critical subspace for the edit. Beyond quantitative gains, our analyses show that SUIT reduces unintended perturbations in hidden states while confining updates to directions that are more effective for editing. Taken together, these findings establish edit-critical subspace identification as a key principle for reliable, low-perturbation knowledge editing. Our code is available at https://github.com/holi-lab/SUIT.

## 1 INTRODUCTION

Large language models (LLMs) retain and recall substantial factual knowledge. However, they often produce incorrect statements due to noisy training data or a temporal shift (Maynez et al., 2020; Ji et al., 2023; Lin et al., 2022). These errors reveal gaps and temporal drift in the model's knowledge, indicating the need for edits to correct these issues. Fine-tuning is commonly used for this purpose, but it is computationally costly and susceptible to overfitting and catastrophic forgetting (Zhang et al., 2024; Bethune et al., 2025; Luo et al., 2025). Consequently, knowledge editing methods have emerged as a promising alternative, enabling targeted edits of specific knowledge while preserving the rest (Yao et al., 2023; Wang et al., 2024b). Among various knowledge editing methods (Wang et al., 2024a), our work builds on the *locate-then-edit* approach. This approach first identifies the weights that store the knowledge, and then directly updates them to apply the edit. It has shown high precision in both mass and sequential editing (Meng et al., 2023b; Fang et al., 2025).

To explain this process, we first describe the standard knowledge editing setup. Given a factual tuple $(e, r, a)$, where $e$ is an entity, $r$ is a relation, and $a$ is an attribute, knowledge editing replaces the old attribute $a$ with a new attribute $a^*$.[1] For example, for the entity $(e,$ *"Chrome"*$)$ and the relation $(r,$ *"was developed by"*$)$, the old attribute $(a,$ *"Google"*$)$ can be edited to the new attribute $(a^*,$ *"Apple"*$)$. Within *locate-then-edit* methods, this editing is performed by viewing the Transformer MLP's down-projection matrix $\mathbf{W}$ as a linear associative memory (Anderson, 1972; Kohonen, 1972; Meng et al., 2023a), where $\mathbf{W}$ maps key vectors to value vectors. In these methods, the key

---

[*]Equal contribution.

[†]Work done while the author was an intern from the Department of Linguistics, Seoul National University.

[‡]Corresponding author.

[1]Much of the knowledge editing literature represents facts in (subject, relation, object) form. However, facts in knowledge editing do not always correspond directly to this form, so we use the more general terms (entity, relation, attribute).

vector $\mathbf{k}$ encodes the entity $e$, and the value vector $\mathbf{v}$ encodes $(r, a)$. The edit is then achieved by computing a new value vector $\mathbf{v}^*$ for the new pair $(r, a^*)$ and redirecting the mapping from $\mathbf{k} \mapsto \mathbf{v}$ to $\mathbf{k} \mapsto \mathbf{v}^*$. Once the pair $(\mathbf{k}, \mathbf{v}^*)$ is specified, the corresponding weight update $\boldsymbol{\Delta}$ to be added to $\mathbf{W}$ can be calculated in closed form such that $(\mathbf{W} + \boldsymbol{\Delta})\mathbf{k} = \mathbf{v}^*$, allowing the weights after knowledge editing to be written as $\mathbf{W}' = \mathbf{W} + \boldsymbol{\Delta}$. Consequently, the result of the edit is fundamentally determined by how $\mathbf{k}$ and $\mathbf{v}^*$ are specified.

An ideal knowledge editing method should edit only the targeted knowledge while preserving unrelated knowledge and thus preventing unintended model perturbation. Accordingly, recent studies have sought to restrict the edited space; a notable example, AlphaEdit (Fang et al., 2025), constrains the edit by enforcing the row space of $\boldsymbol{\Delta}$ to lie in the null space of the key vectors which are computed from existing knowledge that is irrelevant to the specific edit. However, this leaves a critical factor underconstrained: the computation of $\mathbf{k}$ and $\mathbf{v}^*$ themselves. As noted above, the edit outcome is fundamentally determined by how these vectors are specified, and even with constraints on $\boldsymbol{\Delta}$, the high degrees of freedom in these vectors can still lead to unintended perturbation.

To address this, we constrain the computation of $\mathbf{k}$ and $\mathbf{v}^*$ by leveraging the *Linear Representation Hypothesis* (Elhage et al., 2022; Mikolov et al., 2013; Park et al., 2024), which has received limited attention in the knowledge editing literature. This perspective suggests that $\mathbf{k}$ and $\mathbf{v}^*$ are linear combinations of both edit-relevant features (e.g., features related to Chrome's developer attribute) and edit-irrelevant features (e.g., features related to Chrome's lexical form as a proper noun or semantic category as a web browser), and that the computation of these vectors should be restricted to the former. Such a restriction enables us to derive $\mathbf{k}$ and $\mathbf{v}^*$ from narrower, more precise subspaces, which is crucial for preventing unintended model perturbation.

Against this backdrop, we propose **Subspace Knowledge Edit (SUIT)**, which localizes target knowledge to specific subspaces and confines edits within them. Specifically, for the key vector $\mathbf{k}$, we isolate subspaces specific to the entity being edited from those shared across many entities in the context of knowledge editing, such as the property of being a proper noun. For the new value vector $\mathbf{v}^*$, we restrict the update to the subspace that primarily encodes which attribute should be retrieved as the target for the given entity and relation. Across LLaMA3, GPT-J, and Qwen2.5, SUIT substantially outperforms prior methods. In particular, compared with AlphaEdit (Fang et al., 2025), a strong baseline for minimizing knowledge disruption, SUIT achieves much higher specificity (i.e., better preservation of unedited knowledge) and preserves overall model capabilities far more robustly, without sacrificing editing performance. Additionally, our analyses (§ 6) demonstrate how and why SUIT enables precise editing. Specifically, compared to baseline methods, SUIT substantially reduces perturbation at the entity's last token position, where attribute information is predominantly enriched. We further verify that the subspaces and directions we identified are more critical for editing than their complements, and that SUIT successfully isolates its edits to those subspaces.

## 2 RELATED WORK

**Knowledge Editing** Knowledge editing methods can be broadly categorized into three paradigms. *Memory-based approaches* preserve the original model by storing edited knowledge externally and retrieving it during inference (Mitchell et al., 2022b; Hartvigsen et al., 2023). This design avoids direct interference with the base model, but growing edits increase memory and retrieval overhead, limiting scalability. *Meta-learning approaches* train hyper-networks to generate weight updates for efficient editing (Mitchell et al., 2022a; Cao et al., 2021). However, they often generalize poorly to diverse edit distributions or large-scale sequential edits. Finally, *locate-then-edit approaches* identify parameters to update and directly modify model weights, while restricting updates to parameters associated with factual associations in language models (Meng et al., 2023a;b; Fang et al., 2025). This makes them effective for sequential and large-scale editing while preserving overall model performance (Wang et al., 2024b), providing a practical foundation for continual knowledge editing. Accordingly, SUIT is designed as a method in this paradigm.

**Linear Representation Hypothesis** The *Linear Representation Hypothesis* posits that the hidden states of language models are a linear combination of interpretable features, with each feature occupying a distinct subspace (Elhage et al., 2022; Mikolov et al., 2013; Park et al., 2024). Indeed, numerous studies have empirically demonstrated this hypothesis, showing that a variety of features—such as

syntax, position, and factual knowledge, among others—can be identified within specific, decomposable subspaces or directions (Huben et al., 2024; Ji et al., 2023; Huang et al., 2024). We apply this perspective to knowledge editing, decomposing key and value vectors into features relevant and irrelevant to the specific edit and confining edits within the identified subspaces.

## 3 Preliminaries

### 3.1 Linear Associative Memory

Knowledge editing in language models aims to edit a fact triplet from $(e, r, a)$ to $(e, r, a^*)$. The *locate-then-edit* methods (Meng et al., 2023a;b; Fang et al., 2025) achieve this by viewing the MLP's down-projection layer as a linear associative memory that maps keys to values. From this perspective, the MLP's down-projection layer can be expressed as:

$$\mathbf{W}\mathbf{k} = \mathbf{v}.$$

Here, the down-projection matrix $\mathbf{W}$ maps the key vector $\mathbf{k}$, which is the MLP's up-projection activation, to the value vector $\mathbf{v}$. Based on this, the core idea of the *locate-then-edit* methods is that the key vector $\mathbf{k}$ encodes the entity $e$, while the value vector $\mathbf{v}$ encodes the relation $r$ and attribute $a$. To edit the model's knowledge, these methods update the matrix $\mathbf{W}$ by adding an update matrix $\boldsymbol{\Delta}$. This change redirects the mapping of the key vector $\mathbf{k}$ from the value vector $\mathbf{v}$, encoding $(r, a)$, to a new value vector $\mathbf{v}^*$ that encodes $(r, a^*)$. The task is thus to find $\boldsymbol{\Delta}$ such that:

$$(\mathbf{W} + \boldsymbol{\Delta})\mathbf{k} \approx \mathbf{v}^*.$$

Since $\mathbf{W}\mathbf{k} = \mathbf{v}$, this simplifies to $\boldsymbol{\Delta}\mathbf{k} \approx \mathbf{v}^* - \mathbf{v}$. We define $\mathbf{r} := \mathbf{v}^* - \mathbf{v}$ as the residual vector.

Building on this principle, MEMIT (Meng et al., 2023b) extends the approach to edit many facts at once. For a batch of $n$ facts $(e, r, a^*)_i$ where $i = 1, \ldots, n$, it first computes the corresponding key vectors $\mathbf{k}_i$ and residual vectors $\mathbf{r}_i$. These are then concatenated to form the key matrix $\mathbf{K} = [\mathbf{k}_1 \mid \mathbf{k}_2 \mid \cdots \mid \mathbf{k}_n]$ and the residual matrix $\mathbf{R} = [\mathbf{r}_1 \mid \mathbf{r}_2 \mid \cdots \mid \mathbf{r}_n]$. MEMIT then derives a closed-form solution for the update matrix $\boldsymbol{\Delta}$ using these matrices.

Subsequently, AlphaEdit (Fang et al., 2025), based on MEMIT, introduces a new formula for $\boldsymbol{\Delta}$ designed to better preserve the model's existing knowledge. The proposed formula is:

$$\boldsymbol{\Delta} = \mathbf{R}\mathbf{K}^\top \mathbf{P} \left( \mathbf{K}_p \mathbf{K}_p^\top \mathbf{P} + \mathbf{K}\mathbf{K}^\top \mathbf{P} + \mathbf{I} \right)^{-1}. \tag{1}$$

Here, $\mathbf{P}$ is a nullspace projection matrix introduced to preserve the model's existing knowledge, and $\mathbf{K}_p$ aggregates key matrices from prior edits. With $\mathbf{P}$ and $\mathbf{K}_p$ fixed, the computation of $\boldsymbol{\Delta}$ is determined by the **key vector** $\mathbf{k}$ for the entity $e$ and the **residual vector** $\mathbf{r}$ representing the change from $a$ to $a^*$, making accurate estimation of these vectors crucial. Appendix A provides the full derivation of the update formula and the computation details for $\mathbf{P}$ and $\mathbf{K}_p$.

### 3.2 Computing the Key Vector $\mathbf{k}$ and the Residual Vector $\boldsymbol{\delta}$

**Key Vector.** Suppose we want to modify the original fact $(e, r, a)$ into $(e, r, a^*)$. To compute the key vector $\mathbf{k}$, we extract the MLP up-projection activation from the last token of the entity (e.g., *"rome"* in *"Chrome"*). Following Meng et al. (2023a), to improve the generalization of the key vector, we repeat this with various prefixes, and average the extracted MLP up-projection activations to obtain the final key vector $\mathbf{k}$.

**Residual Vector.** While ROME (Meng et al., 2023a) targeted a single layer, most subsequent approaches perform edits across multiple layers. These multi-layer methods do not compute the residual vector $\mathbf{r}$ separately for each layer. Instead, they first calculate an *entire residual vector* $\boldsymbol{\delta}$ (henceforth, the residual vector) from the residual stream of the final layer being edited. This vector $\boldsymbol{\delta}$ is then distributed proportionally to determine the specific $\mathbf{r}$ for each layer involved in the edit. To obtain $\boldsymbol{\delta}$, it is added to the residual stream $\mathbf{h}$ at the entity's last token position in the last modified layer and optimized via gradient descent to maximize the logit of the new attribute $a^*$ (*"Apple"*) given the input $e$ and $r$ (*"Chrome was developed by"*). To prevent overfitting to the new fact, which can lead to

the corruption of unrelated knowledge, a regularization term $\mathcal{R}$ is added to the loss function (details in Appendix A.4). The optimization objective is formulated as:

$$\boldsymbol{\delta} = \arg\min_{\tilde{\boldsymbol{\delta}}} \left\{ -\log p\Big(a^* \mid \mathbf{h}^* \leftarrow \mathbf{h} + \tilde{\boldsymbol{\delta}}\Big) + \mathcal{R} \right\}. \tag{2}$$

## 4    SUIT: SUBSPACE KNOWLEDGE EDIT

### 4.1    KNOWLEDGE EDITING UNDER LINEAR REPRESENTATION HYPOTHESIS

According to the *Linear Representation Hypothesis*, the key and value vectors within an MLP's down-projection layer can be viewed as a composition of interpretable features. For knowledge editing, we hypothesize that these vectors consist of features that are either relevant or irrelevant to the specific edit. The key vector $\mathbf{k}$, which encodes the entity, can be divided into entity-specific features and more general, entity-agnostic features that activate similarly across many entities. Similarly, the value vector $\mathbf{v}$, encoding the relation and attribute, contains features that primarily define the attribute ($a$ or $a^*$) alongside other less relevant features.

To ensure that knowledge edits are precise—modifying only the target knowledge while preserving other knowledge—we propose introducing explicit constraints. When computing the key vector $\mathbf{k}$, we aim to consider only the subspace occupied by its entity-specific features. Likewise, when computing the residual vector $\boldsymbol{\delta}$, we aim to consider only the feature directions that significantly influence the attribute's logit. Accordingly, in the following sections we obtain the subspace-aware key vector $\mathbf{k}'$ (§ 4.2) and the subspace-aware residual vector $\boldsymbol{\delta}'$ (§ 4.3); using these, we compute the update matrix $\boldsymbol{\Delta}$ via Eq. (1).

### 4.2    SUBSPACE-AWARE COMPUTATION FOR OUR KEY VECTOR $\mathbf{k}'$

When computing our key vector $\mathbf{k}'$ that encodes the entity $e$, we consider two complementary subspaces of the key vector space: the *entity-specific* subspace $\mathcal{K}_s$, in which key vector components exhibit high variance across different entities, and the *entity-agnostic* subspace $\mathcal{K}_s^\perp$, in which key vector components exhibit low variance. Since components in $\mathcal{K}_s^\perp$ activate similarly across many entities, retaining them in $\mathbf{k}$ can cause the edit to affect unrelated entities. Accordingly, our objective is to isolate the entity-specific component of $\mathbf{k}$ that lies within $\mathcal{K}_s$.

We begin with the key vector $\mathbf{k}$, as defined in § 3.2. We then decompose this vector into its entity-specific component, $\mathbf{k}_s \in \mathcal{K}_s$, and its entity-agnostic component, $\mathbf{k}_{\sim s} \in \mathcal{K}_s^\perp$. Our key vector $\mathbf{k}'$ is obtained by removing this entity-agnostic component:

$$\mathbf{k}' = \mathbf{k} - \mathbf{k}_{\sim s} = \mathbf{k}_s.$$

The core task is to identify $\mathcal{K}_s^\perp$ and compute these vector components. We accomplish this through the following procedure.

We sample $N = 10{,}000$ entities from PARAREL (Elazar et al., 2021), a dataset of $(e, r, a)$ triplets derived from Wikidata. For each entity, we compute its key vector $\mathbf{k}$ to form the matrix $\mathbf{K}_{\text{entity}} = [\mathbf{k}_1 \mid \mathbf{k}_2 \mid \cdots \mid \mathbf{k}_{10000}]$.

Applying singular value decomposition (SVD) to this matrix yields:

$$\mathbf{K}_{\text{entity}} = \mathbf{U}\,\mathbf{S}\,\mathbf{V}^\top,$$

where $\mathbf{S} = \operatorname{diag}(\sigma_1, \sigma_2, \ldots, \sigma_r)$ with $\sigma_1 \geq \sigma_2 \geq \cdots \geq \sigma_r$ denotes the singular values, and $\mathbf{U} = [\mathbf{u}_1 \mid \mathbf{u}_2 \mid \cdots \mid \mathbf{u}_r]$ contains the corresponding left singular vectors.

To determine how many critical components to remove, we introduce a hyperparameter $\tau_{\text{energy}} \in [0, 1)$ that represents the proportion of total variance (energy) to isolate. Let $E_{\text{total}} = \sum_{i=1}^{r} \sigma_i^2$ be the total energy of $\mathbf{K}_{\text{entity}}$. We find the smallest integer $m$ such that the cumulative energy of the first $m$ components reaches or exceeds this threshold: $\sum_{i=1}^{m} \sigma_i^2 \geq \tau_{\text{energy}} \cdot E_{\text{total}}$. Let $\mathbf{U}_{\sim s} = [\mathbf{u}_1 \mid \cdots \mid \mathbf{u}_m]$ denote the matrix of the first $m$ left singular vectors. We then define the entity-agnostic subspace $\mathcal{K}_s^\perp$ as their span:

$$\mathcal{K}_s^\perp := \operatorname{span}(\mathbf{U}_{\sim s}).$$

Finally, we use the matrix $\mathbf{U}_{\sim s}\mathbf{U}_{\sim s}^{\top}$ to project the key vector $\mathbf{k}$ onto the entity-agnostic subspace $\mathcal{K}_s^{\perp}$. By subtracting this projection $\mathbf{k}_{\sim s}$, we remove the entity-agnostic component, leaving only the entity-specific component $\mathbf{k}_s$. This procedure yields the constrained key vector $\mathbf{k}'$, which now contains only the *features* relevant to the entity being edited:

$$\mathbf{k}' = \mathbf{k} - \mathbf{k}_{\sim s}, \quad \mathbf{k}_{\sim s} = \mathbf{U}_{\sim s}\mathbf{U}_{\sim s}^{\top}\mathbf{k}.$$

### 4.3 Subspace-Aware Computation for Our Residual Vector $\boldsymbol{\delta}'$

The residual vector $\boldsymbol{\delta}$ is computed to modify the residual stream $\mathbf{h}$ so that it encodes $(r, a^*)$ (see § 3.2). Rather than altering the full-dimensional residual stream $\mathbf{h}$, our approach is to target a low-dimensional subspace that governs the model's prediction for the given $(e, r)$ pair.

We hypothesize that this targeted modification can be achieved within a two-dimensional subspace spanned by two critical unit feature directions, $\mathbf{w}_1$ and $\mathbf{w}_2$, associated with $a$ and $a^*$. Specifically, increasing the magnitude of $\mathbf{h}$ along $\mathbf{w}_1$ (i.e., $\mathbf{h}^{\top}\mathbf{w}_1$) raises the logit of the new attribute $a^*$, while decreasing its magnitude along $\mathbf{w}_2$ (i.e., $\mathbf{h}^{\top}\mathbf{w}_2$) suppresses the logit of the old attribute $a$. Our goal is to identify $\mathbf{w}_1$ and $\mathbf{w}_2$ and swap these directional magnitudes of $\mathbf{h}$, thereby increasing the logit of $a^*$ toward the original level of $a$ and decreasing the logit of $a$ toward the original level of $a^*$. For simplicity, we ignore interactions between the two directions and implement the edit as a simple additive update (details in Appendix B). The updated residual stream $\mathbf{h}^*$ is:

$$\mathbf{h}^* = \mathbf{h} + \boldsymbol{\delta}', \quad \boldsymbol{\delta}' = (\mathbf{h}^{\top}\mathbf{w}_2 - \mathbf{h}^{\top}\mathbf{w}_1)\mathbf{w}_1 + (\mathbf{h}^{\top}\mathbf{w}_1 - \mathbf{h}^{\top}\mathbf{w}_2)\mathbf{w}_2.$$

The process for finding the optimal basis vectors, $\{\mathbf{w}_1, \mathbf{w}_2\}$, follows a similar structure to the optimization for the residual vector $\boldsymbol{\delta}$ shown in Eq. (2). The primary objective is to maximize the logit of the new attribute $a^*$. While Eq. (2) included a general regularization term $\mathcal{R}$, it is unnecessary in our approach as we constrain the update to a two-dimensional subspace. To encourage the basis vectors $\mathbf{w}_1$ and $\mathbf{w}_2$ to represent distinct directions, we introduce a directional penalty term, formulated as $(\hat{\mathbf{w}}_1^{\top}\hat{\mathbf{w}}_2)^2$. The complete optimization objective reflecting this is therefore formulated as:

$$\{\mathbf{w}_1, \mathbf{w}_2\} = \arg\min_{\hat{\mathbf{w}}_1, \hat{\mathbf{w}}_2}\left\{-\log p\left(a^* \mid \mathbf{h}^* \leftarrow \mathbf{h} + \hat{\boldsymbol{\delta}}'\right) + \lambda\left(\hat{\mathbf{w}}_1^{\top}\hat{\mathbf{w}}_2\right)^2\right\},$$

where the hyperparameter $\lambda$ is the penalty weight that controls the strength of the directional penalty.

## 5 Experiments

This section describes the experimental setup (§ 5.1), presents the main results in Tables 1 and 2, and reports additional experiments in § 5.3.

### 5.1 Experimental Setup

**Models, Baselines, and Datasets.** We conduct our experiments on LLaMA3-Instruct (8B) (Grattafiori et al., 2024), GPT-J (6B) (Wang & Komatsuzaki, 2021), and Qwen2.5-Instruct (7B) (Yang et al., 2024). We compare SUIT against several representative model editing baselines, including Fine-Tuning (FT-L) (Zhu et al., 2020), MEND (Mitchell et al., 2022a), PMET (Li et al., 2024), MEMIT (Meng et al., 2023b), and AlphaEdit (Fang et al., 2025). Due to space constraints, we defer results for additional baselines (FT-W (Zhu et al., 2020), ROME (Meng et al., 2023a), RECT (Gu et al., 2024), PRUNE (Ma et al., 2024), NSE (Jiang et al., 2024)) to Appendix E.1. We also provide the hyperparameters used for all methods, including SUIT, in Appendix C.2, and describe in § 6.3 how the SUIT-specific hyperparameters $\tau_{\text{energy}}$ and $\lambda$ are selected. We use two widely used knowledge editing benchmarks: COUNTERFACT (Meng et al., 2023a) and zsRE (Levy et al., 2017).

**Evaluation Metrics and Setup.** We report three standard knowledge editing metrics: *Efficacy*, *Generalization*, and *Specificity*. *Efficacy* (*Eff.*) assesses whether the model generates the new attribute $a^*$ for a given *rewrite prompt* $(e, r)$, while *Generalization* (*Gen.*) measures the same for *paraphrase prompts*. *Specificity* (*Spe.*) checks whether an edit causes unintended changes to other knowledge, using *neighborhood prompts*. For COUNTERFACT, we also report the generation-quality metrics *Consistency* (*Con.*) and *Fluency* (*Flu.*). Further details are provided in Appendix D.

Table 1: COUNTERFACT results for sequentially editing 1,000 facts with different batch sizes (1, 10, and 100); *GC*, *Flu.*, and *Con.* are reported for batch size 100. Best numbers are **bold**; second-best are underlined. Abbreviations: *Eff.* = Efficacy, *Gen.* = Generalization, *Spe.* = Specificity, *GC* = General Capability, *Flu.* = Fluency, *Con.* = Consistency.

| | Method | batch size = 1 | | | | batch size = 10 | | | | batch size = 100 | | | | | | |
|---|---|---|---|---|---|---|---|---|---|---|---|---|---|---|---|---|
| | | $S$ ↑ | *Eff.* ↑ | *Gen.* ↑ | *Spe.* ↑ | $S$ ↑ | *Eff.* ↑ | *Gen.* ↑ | *Spe.* ↑ | $S$ ↑ | *Eff.* ↑ | *Gen.* ↑ | *Spe.* ↑ | *GC* ↑ | *Flu.* ↑ | *Con.* ↑ |
| **LLaMA3** | Pre-edit | 0.0 | 0.0 | 0.0 | 100.0 | 0.0 | 0.0 | 0.0 | 100.0 | 0.0 | 0.0 | 0.0 | 100.0 | 63.4 | 634.9 | 20.9 |
| | FT-L | 0.5 | 3.9 | 2.1 | 0.2 | 0.0 | 9.4 | 3.8 | 0.0 | 1.9 | 9.9 | 1.4 | 1.3 | 6.2 | 438.5 | 19.2 |
| | MEND | 0.0 | 0.0 | 0.0 | 0.0 | 0.0 | 0.0 | 0.0 | 0.0 | 0.0 | 0.0 | 0.0 | 1.3 | 0.0 | 519.5 | 0.6 |
| | MEMIT | 0.0 | 0.0 | 0.0 | 0.0 | 0.0 | 0.0 | 0.0 | 0.0 | 48.3 | 76.2 | 74.0 | 28.2 | 60.8 | 628.9 | 36.7 |
| | PMET | 0.0 | 0.0 | 0.0 | 0.0 | 0.0 | 0.0 | 0.0 | 0.0 | 37.6 | 56.6 | 56.0 | 22.6 | 50.1 | 608.8 | 33.8 |
| | AlphaEdit | 36.6 | **96.5** | **90.2** | 16.5 | 37.4 | **96.5** | **90.4** | 17.0 | 55.8 | 97.3 | 88.7 | 31.0 | 62.2 | **633.6** | **38.6** |
| | SUIT | **85.4** | **99.6** | 89.5 | **71.9** | **85.6** | **99.9** | 89.8 | **71.9** | **86.8** | **99.7** | **90.3** | **74.2** | **63.0** | 631.2 | 38.2 |
| **GPT-J** | Pre-edit | 0.0 | 0.0 | 0.0 | 100.0 | 0.0 | 0.0 | 0.0 | 100.0 | 0.0 | 0.0 | 0.0 | 100.0 | 24.3 | 621.1 | 23.9 |
| | FT-L | 12.5 | 31.9 | 23.4 | 6.0 | 9.3 | 47.8 | 32.8 | 3.7 | 13.3 | 64.9 | 46.7 | 5.3 | **24.2** | 334.2 | 12.2 |
| | MEND | 0.0 | 0.0 | 0.0 | 0.0 | 0.0 | 0.0 | 0.0 | 0.0 | 0.0 | 0.0 | 0.0 | 0.0 | 0.0 | 515.0 | 2.7 |
| | MEMIT | 0.0 | 0.0 | 0.0 | 0.0 | 46.1 | 75.1 | 74.1 | 26.1 | 60.2 | 92.0 | 90.4 | 35.8 | 20.1 | 617.4 | 48.5 |
| | PMET | 0.0 | 0.0 | 0.0 | 0.1 | 0.0 | 0.0 | 0.0 | 0.0 | 57.4 | 84.6 | 84.5 | 35.0 | 19.1 | 618.5 | 44.8 |
| | AlphaEdit | 72.8 | **98.9** | **94.8** | 48.7 | 76.3 | **99.2** | **96.0** | 53.1 | 73.0 | 98.3 | **95.0** | 49.0 | 19.5 | **621.8** | **49.9** |
| | SUIT | **82.3** | 98.4 | 92.7 | **64.5** | **84.3** | 99.1 | 94.3 | **67.2** | **84.7** | **99.2** | 94.6 | **67.7** | 20.4 | 619.4 | 49.4 |
| **Qwen2.5** | Pre-edit | 0.0 | 0.0 | 0.0 | 100.0 | 0.0 | 0.0 | 0.0 | 100.0 | 0.0 | 0.0 | 0.0 | 100.0 | 28.9 | 625.5 | 21.9 |
| | FT-L | 11.6 | 21.6 | 19.5 | 6.2 | 12.0 | 28.1 | 22.5 | 5.9 | 10.3 | 47.9 | 31.7 | 4.2 | 0.0 | 476.7 | 4.5 |
| | MEND | 0.0 | 0.0 | 0.0 | 0.0 | 0.0 | 0.0 | 0.0 | 0.0 | 0.0 | 0.0 | 0.0 | 0.0 | 0.0 | 466.3 | 0.1 |
| | MEMIT | 0.0 | 0.0 | 0.0 | 0.0 | 0.0 | 0.1 | 0.0 | 0.0 | 22.6 | 83.0 | 84.0 | 9.2 | **34.8** | 622.1 | 37.0 |
| | PMET | 0.0 | 0.0 | 0.0 | 0.0 | 0.0 | 0.0 | 0.0 | 0.0 | 32.6 | 67.5 | 65.9 | 16.1 | 14.8 | 545.2 | 27.5 |
| | AlphaEdit | 65.0 | **96.2** | **92.4** | 40.1 | 66.6 | **96.0** | **93.2** | 41.9 | 67.8 | 97.1 | **91.6** | 43.4 | 28.1 | **626.2** | **41.0** |
| | SUIT | **85.6** | **99.3** | 90.3 | **72.0** | **85.7** | **99.3** | 90.8 | **71.9** | **86.1** | **99.2** | **91.6** | **72.3** | 30.8 | **626.2** | 37.4 |

To faithfully measure the effects of knowledge editing, especially the preservation of existing knowledge, we focus on instances where the model already produced the target answer before editing. For COUNTERFACT, we keep only facts for which the pre-edit model generates the ground-truth answer. This ensures that *Eff.* and *Gen.* measure edits to known facts rather than insertion of facts the model did not know before. Under the same selection, *Spe.* measures whether knowledge previously exhibited by the model remains preserved after editing. Following prior work (Meng et al., 2023a;b; Fang et al., 2025), we also report $S$, the harmonic mean of *Eff.*, *Gen.*, and *Spe.*, which summarizes how well a method balances these metrics under their trade-offs. For ZSRE, however, restricting evaluation to pre-edit-known instances is not feasible, because the pre-edit model rarely generates the ground-truth answer as annotated in the dataset (see Appendix D.3 for quantitative details and rationale). For *Spe.*, we therefore use the model's own pre-edit generated response as the reference so that the metric reflects preservation of previously exhibited knowledge. We then evaluate whether the first 1–4 tokens of that response are exactly preserved after editing.

**General Capability.** We evaluate the model's General Capability (*GC*) to assess whether knowledge editing degrades performance on general tasks. The *GC* score is measured after sequentially editing 1,000 facts with batch size 100. It is computed as the average F1 score across six benchmarks: MMLU (Hendrycks et al., 2021) and five GLUE tasks (Wang et al., 2019) (NLI, MRPC, SST, RTE, and CoLA). Detailed per-benchmark results are provided in Appendix E.2.

## 5.2 EXPERIMENTAL RESULTS

Table 1 shows that on COUNTERFACT, SUIT most consistently achieves the best overall balance between editing success (*Eff.* and *Gen.*) and preservation (*Spe.*) across LLaMA3, GPT-J, and Qwen2.5. This is also reflected in the harmonic mean score $S$, where SUIT is highest in all settings. Importantly, SUIT does not obtain this advantage by sacrificing editing performance; rather, it maintains strong *Eff.* and *Gen.* while also achieving substantially higher *Spe.*. The batch size settings denote how many facts are edited in each update step, while keeping the total number of edited facts fixed at 1,000. Therefore, smaller batch sizes require more update steps and thus more weight updates to complete the same number of edits. Under smaller batch sizes, several baselines collapse to near-zero performance, indicating model collapse under frequent cumulative weight updates. In contrast, SUIT maintains strong performance even in these harder settings and, compared with AlphaEdit, shows much less degradation in *Spe.* as batch size decreases. SUIT also preserves *Flu.*, *Con.*, and *GC*, showing that it can edit knowledge effectively without degrading overall model performance. Table 2

Table 2: zSRE results for sequential editing of 1000 facts with a batch size of 100. *Specificity* measures the exact preservation of the first $\ell$ tokens of the pre-edit response.

| Method | LLaMA3 | | | | | | GPT-J | | | | | | Qwen2.5 | | | | | |
|---|---|---|---|---|---|---|---|---|---|---|---|---|---|---|---|---|---|---|
| | *Eff.* ↑ | *Gen.* ↑ | *Spe.* ↑ | | | | *Eff.* ↑ | *Gen.* ↑ | *Spe.* ↑ | | | | *Eff.* ↑ | *Gen.* ↑ | *Spe.* ↑ | | | |
| | | | $\ell=1$ | $\ell=2$ | $\ell=3$ | $\ell=4$ | | | $\ell=1$ | $\ell=2$ | $\ell=3$ | $\ell=4$ | | | $\ell=1$ | $\ell=2$ | $\ell=3$ | $\ell=4$ |
| Pre-edit | 35.9 | 34.8 | 100.0 | 100.0 | 100.0 | 100.0 | 27.2 | 26.3 | 100.0 | 100.0 | 100.0 | 100.0 | 34.4 | 33.8 | 100.0 | 100.0 | 100.0 | 100.0 |
| FT-L | 34.8 | 34.0 | 0.0 | 0.0 | 0.0 | 0.0 | 69.3 | 60.6 | 0.0 | 0.0 | 0.0 | 0.0 | 12.5 | 11.7 | 0.0 | 0.0 | 0.0 | 0.0 |
| MEND | 0.0 | 0.0 | 0.0 | 0.0 | 0.0 | 0.0 | 0.4 | 0.4 | 0.0 | 0.0 | 0.0 | 0.0 | 0.0 | 0.0 | 0.0 | 0.0 | 0.0 | 0.0 |
| MEMIT | 89.3 | 85.7 | 30.1 | 22.9 | 21.1 | 19.2 | 96.8 | 90.8 | 77.9 | 77.7 | 30.8 | 23.2 | 96.2 | 91.4 | 15.7 | 11.8 | 10.6 | 9.3 |
| PMET | 89.2 | 83.9 | 26.7 | 21.3 | 18.9 | 16.7 | 95.1 | 89.2 | 67.4 | 67.0 | 23.3 | 17.1 | 82.1 | 76.3 | 5.8 | 2.1 | 1.5 | 1.1 |
| AlphaEdit | 93.5 | 88.7 | 31.2 | 22.1 | 19.8 | 18.3 | 99.7 | 97.9 | 94.4 | 94.1 | 50.5 | 41.6 | 94.5 | 89.8 | 30.1 | 25.0 | 22.8 | 19.5 |
| SUIT | 95.2 | 85.7 | 51.9 | 43.2 | 41.2 | 39.8 | 99.7 | 94.3 | 99.3 | 99.1 | 70.4 | 61.6 | 99.5 | 92.2 | 64.4 | 57.3 | 54.5 | 51.7 |

shows that the same pattern observed in Table 1 also holds on zSRE. Across LLaMA3, GPT-J, and Qwen2.5, and across token-length settings $\ell$, SUIT consistently attains substantially higher *Spe.* than prior methods, while retaining strong editing success (*Eff.* and *Gen.*).

## 5.3 ADDITIONAL EXPERIMENTS

We present additional experiments to further evaluate SUIT along several dimensions.

- **Stability under extended sequential editing.** We extend the evaluation to 5,000 edits and track *GC* every 100 edits (Fig. 1). As edits accumulate, SUIT remains substantially more robust than the strong baseline AlphaEdit, while Appendix E.3 confirms that SUIT also maintains high editing performance after 5,000 edits.
- **Scaling to a larger model.** To test whether SUIT's advantages persist at larger scale, we evaluate LLaMA2-13B (Touvron et al., 2023) on COUNTERFACT. SUIT preserves a strong editing–preservation balance and achieves the highest $S$ among the baselines; see Appendix E.4 for details.

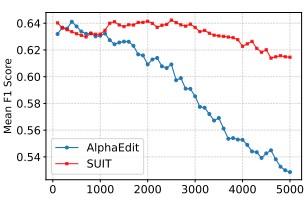

Figure 1: *GC* over 5,000 sequential edits.

- **Context robustness of edited facts.** While SUIT has been shown to preserve existing knowledge and maintain strong editing performance when prompted with entity-relation pairs, precise edits should also be robust even when these pairs appear after various contexts in the prompt. We evaluate this with CHED (Park et al., 2025), which measures the correctness of edited facts under diverse prefix contexts. As shown in Appendix E.5, SUIT consistently outperforms baselines, demonstrating robustness to context variation.
- **Ripple effects of edited facts.** Effective model editing should consistently reflect an edit's implications on logically related knowledge. To evaluate this, we employ RIPPLEEDITS (Cohen et al., 2024), which measures whether an edit correctly propagates to associated facts across multiple logical relations. As shown in Appendix E.6, SUIT is competitive with or superior to existing baselines, demonstrating its ability to maintain logical consistency.

## 6 ANALYSIS

To further investigate SUIT, we conducted a series of analyses. We demonstrate its effectiveness in reducing perturbation at the entity's last token. We then empirically validate our hypotheses regarding the subspaces identified for computing the key and residual vectors. Finally, we present a hyperparameter analysis and an ablation study. All analyses were performed using LLaMA3. For experiments requiring edited models, we used models edited with 10 batches of 100 edits each from COUNTERFACT.

### 6.1 REDUCING ENTITY'S LAST TOKEN PERTURBATION

Ideally, an edited model is expected to behave identically to the original model on unedited knowledge, without introducing any perturbation. Prior research (Meng et al., 2023a; Geva et al., 2023; Chughtai et al., 2024) has demonstrated that an entity's attributes are predominantly enriched at the last token position of the entity. Accordingly, *locate-then-edit* methods perform edits by deriving key and value

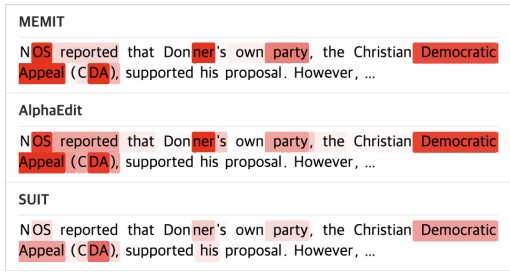

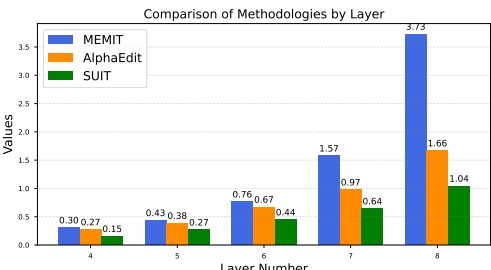

Figure 2: Comparison of token-level perturbations in residual streams.

Figure 3: Norm differences of MLP outputs at the entity's last token position across methods.

vectors directly from this position. This approach, however, induces significant perturbation precisely at this critical location.

Fig. 2 visualizes how much the residual stream changes in the final edited layer by comparing the pre-edit model with models edited using MEMIT, AlphaEdit, and SUIT. Specifically, we compute the $L_2$ norm of the difference between the residual streams of the original and each edited model, which measures the magnitude of the perturbation introduced by editing at each token. These values are shown as color intensity over a sample paragraph from the Wikinews Article Dataset, where darker colors indicate larger changes (i.e., greater perturbation magnitudes) relative to the pre-edit model. The figure demonstrates that, compared to other tokens, all methods generally exhibit a more notable perturbation at the last token of the entity (e.g., *"OS"* in *"NOS"*, *"ner"* in *"Donner"*). Notably, among these methods, the perturbation from SUIT is visibly less pronounced than that of MEMIT and AlphaEdit. Further examples confirming this pattern are provided in Appendix J.

To quantify the perturbation from editing, we analyze the output of the MLP layers, which are the direct targets of the edits. Using unedited knowledge from COUNTERFACT dataset, we measure the $L_2$ distance between the outputs of the original and edited models at the entity's last token position. As shown in Fig. 3, while all three methods induce a non-zero perturbation, SUIT consistently exhibits the smallest change. This indicates that SUIT performs more localized and precise edits, affecting unrelated knowledge the least.

## 6.2 ANALYSIS FOR SUBSPACES

In this section, we investigate whether the subspaces $\mathcal{K}_s$, $\mathcal{K}_s^\perp$ and the directions $\mathbf{w}_1$, $\mathbf{w}_2$, which we identified to find the key vector $\mathbf{k}$ and the residual vector $\boldsymbol{\delta}$, indeed correspond to the feature subspaces we hypothesized. We hypothesized that $\mathcal{K}_s$ is an entity-specific feature subspace that activates differently for each entity. Conversely, its orthogonal subspace $\mathcal{K}_s^\perp$ is assumed to be an entity-agnostic feature space, activating similarly across different entities. Furthermore, we posited that $\mathbf{w}_1$ and $\mathbf{w}_2$ are crucial directions for the update. Specifically, when their scaled versions are added to the residual stream $\mathbf{h}$, we believe $\mathbf{w}_1$ is the principal direction for increasing the logit of the new attribute $a^*$, while $\mathbf{w}_2$ is the principal direction for decreasing the logit of the old attribute $a$.

### 6.2.1 ANALYSIS FOR $\mathcal{K}_s$ AND $\mathcal{K}_s^\perp$

To investigate whether the subspaces identified via PARAREL generalize to other datasets, we designed an experiment to measure component variance across different datasets. We extracted key vectors $\mathbf{k}$ for 5,000 randomly selected entities from COUNTERFACT and ZSRE datasets. Each key vector $\mathbf{k}$ was then decomposed into its component $\mathbf{k}_s$ in $\mathcal{K}_s$ and its component $\mathbf{k}_{\sim s}$ in $\mathcal{K}_s^\perp$ using the subspace derived from PARAREL. Finally, we computed the variance for each set of components across all 5,000 entities to verify whether the components in $\mathcal{K}_s$ vary across entities, while the components in $\mathcal{K}_s^\perp$ remain relatively similar across different entities, even on other datasets.

The results of our variance analysis, presented in Table 3, align with our initial expectations. Across both datasets, the variance of the entity-specific components $\mathbf{k}_s$ was significantly higher than that of the entity-agnostic ones $\mathbf{k}_{\sim s}$, being approximately 2.6 times higher for COUNTERFACT and 4.5 times higher for ZSRE. This finding is consistent with our hypothesis, suggesting that $\mathcal{K}_s$ may

Table 3: Variance of decomposed key vector components across 5,000 entities.

|  | COUNTERFACT | ZSRE |
|---|---|---|
| $V(\mathbf{k}_{\sim s})$ | 2.041 | 1.333 |
| $V(\mathbf{k}_s)$ | 5.269 | 5.938 |

Table 4: Proportion of the modification affecting entity-agnostic components.

| Prompt type | MEMIT | AlphaEdit | SUIT |
|---|---|---|---|
| rewrite | 0.2814 | 0.4623 | 0.0035 |
| paraphrase | 0.2929 | 0.4717 | 0.0039 |
| neighborhood | 0.6805 | 0.8114 | 0.0201 |

capture entity-specific features that vary across individuals, while $\mathcal{K}_s^{\perp}$ appears to contain more stable, entity-agnostic information.

Furthermore, we conducted an experiment to verify that our update matrix, $\boldsymbol{\Delta}$, interacts less with the subspace $\mathcal{K}_s^{\perp}$. To test this, we compared our method against MEMIT and AlphaEdit on the *rewrite*, *paraphrase*, and *neighborhood prompts* (defined in § 5.1) in COUNTERFACT. Specifically, we measured the proportion of the update that affects the entity-agnostic components, calculated as $\|\boldsymbol{\Delta}\mathbf{k}_{\sim s}\|^2/\|\boldsymbol{\Delta}\mathbf{k}\|^2$. The results computed from the update matrix $\boldsymbol{\Delta}$ at the first edited layer are presented in Table 4. The results clearly demonstrate that for all prompt types, the proportion of the update affecting the entity-agnostic space $\|\boldsymbol{\Delta}\mathbf{k}_{\sim s}\|^2/\|\boldsymbol{\Delta}\mathbf{k}\|^2$ is negligible for our method, SUIT. This is in stark contrast to MEMIT and AlphaEdit, which show a significantly higher proportion of their modifications impacting these common components. This finding suggests that SUIT successfully isolates its edits to the entity-specific space $\mathcal{K}_s$, leaving the shared, entity-agnostic knowledge largely untouched. We can, therefore, infer that this precise targeting is a key reason for SUIT's enhanced specificity for neighborhood prompts.

### 6.2.2 ANALYSIS FOR $\mathbf{w}_1$ AND $\mathbf{w}_2$

Next, we verified our hypothesis that the subspace spanned by $\mathbf{w}_1$ and $\mathbf{w}_2$ represents the critical directions for increasing the logit of the new attribute $a^*$. To do this, we first computed the residual vector $\boldsymbol{\delta}$ using the original method (§ 3.2). We then decomposed this vector into two distinct components: the component lying in the subspace $\mathrm{span}(\mathbf{w}_1, \mathbf{w}_2)$, $\boldsymbol{\delta}_{\|W}$, and the remaining orthogonal component, $\boldsymbol{\delta}_{\perp W}$. The projection is calculated as:

$$\boldsymbol{\delta}_{\|W} = P_W\boldsymbol{\delta}, \quad \text{where } W = [\mathbf{w}_1, \ \mathbf{w}_2], \quad P_W = W\big(W^TW\big)^{-1}W^T.$$

To compare the respective effects of these two components on increasing the logit of $a^*$, we added each of $\boldsymbol{\delta}_{\|W}$ and $\boldsymbol{\delta}_{\perp W}$ to the residual stream $\mathbf{h}$ and measured the logit and probability of $a^*$, given the entity $e$ and relation $r$, averaged over the 1,000 edits in COUNTERFACT. The results in Table 5 provide direct evidence for our design choice: constructing updates in the critical subspace $\mathrm{span}(\mathbf{w}_1, \mathbf{w}_2)$ yields a more effective steering signal than using the remaining directions. Although $\boldsymbol{\delta}_{\|W}$ accounts for

Table 5: Results of steering with decomposed components of $\boldsymbol{\delta}$.

| Space | $\boldsymbol{\delta}_{\|W}$ | $\boldsymbol{\delta}_{\perp W}$ |
|---|---|---|
| $\|\boldsymbol{\delta}_{\|/\perp W}\|^2/\|\boldsymbol{\delta}\|^2$ | 24.17 | 75.83 |
| $p(a^*)$ | 0.67 | 0.59 |
| $\mathrm{logit}(a^*)$ | -1.44 | -1.72 |

only about 24% of the total squared norm of $\boldsymbol{\delta}$, it is more effective at increasing the logit and the probability of $a^*$ than the remaining 76% in $\boldsymbol{\delta}_{\perp W}$. This indicates that $\boldsymbol{\delta}$ contains both directions that are critical for increasing the logit of $a^*$ and other less-essential components. These findings support our approach of constructing the residual update from the critical subspace, enabling a more focused and potent update within a much narrower directional space.

We further analyzed the individual roles of $\mathbf{w}_1$ and $\mathbf{w}_2$ to test whether $\mathbf{w}_1$ primarily increases the logit of the new attribute $a^*$ while $\mathbf{w}_2$ primarily decreases the logit of the old attribute $a$. To do this, we decomposed our update vector $\boldsymbol{\delta}'$ into its components along these directions: $\Delta\mathbf{w}_1 = (\mathbf{h}^\top\mathbf{w}_2 - \mathbf{h}^\top\mathbf{w}_1)\mathbf{w}_1$ and $\Delta\mathbf{w}_2 = (\mathbf{h}^\top\mathbf{w}_1 - \mathbf{h}^\top\mathbf{w}_2)\mathbf{w}_2$. We then incrementally added each component to the residual stream $\mathbf{h}$ by scaling it with an interpolation factor $k \in [0, 1]$, and observed the logits for both attributes.

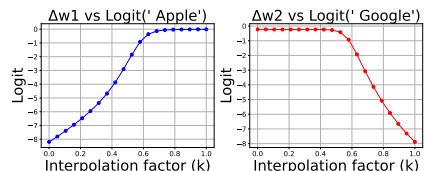

Figure 4: Effects of $\Delta\mathbf{w}_1$ and $\Delta\mathbf{w}_2$ on the logits of "Apple" and "Google"

Fig. 4 shows the results for the edit (*"Chrome"*, *"was developed by"*, *"Apple"*). As expected, the $\Delta\mathbf{w}_1$ component effectively increases the target $a^*$ (*"Apple"*) logit, while the $\Delta\mathbf{w}_2$ component effectively decreases the original $a$ (*"Google"*) logit.

Although $\mathbf{w}_1$ and $\mathbf{w}_2$ drive logits in the expected directions and were trained to play different roles, we find that $\mathbf{w}_1$ also suppresses the old attribute $a$, and $\mathbf{w}_2$ also promotes the new attribute $a^*$, rather than each playing only a single role. The effectiveness of fully disentangling these roles would be worth exploring in future work. For a detailed visualization and a full breakdown of these component effects, please see Appendix F.

### 6.3 HYPERPARAMETER ANALYSIS AND ABLATION STUDY

We analyze the effects of the two hyperparameters in our method: the energy threshold $\tau_{\text{energy}}$, which controls removal of the shared component from $\mathbf{k}$ to obtain $\mathbf{k}'$ (§ 4.2), and the penalty weight $\lambda$, which controls how strongly $\mathbf{w}_1$ and $\mathbf{w}_2$ are encouraged to be distinct when computing $\boldsymbol{\delta}'$ (§ 4.3). We vary one while fixing the other and measure changes in editing metrics. Additionally, we perform separate ablations on $\mathbf{k}'$ and $\boldsymbol{\delta}'$ to isolate their contributions.

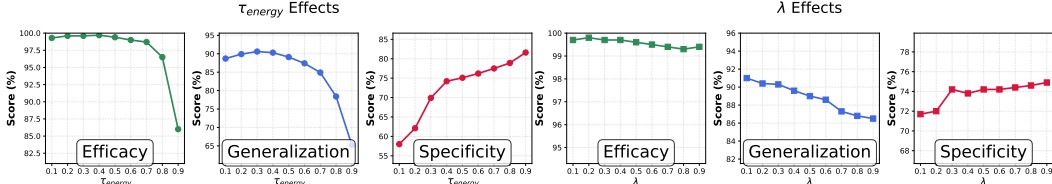

Figure 5: Analysis for hyperparameters $\tau_{\text{energy}}$ and $\lambda$.

**Hyperparameter Analysis.** Fig. 5 shows how $\tau_{\text{energy}}$ and $\lambda$ affect editing metrics on COUNTER-FACT (LLaMA3, 1,000 edits, batch size 100). We vary one hyperparameter while fixing the other ($\lambda = 0.3$ or $\tau_{\text{energy}} = 0.4$). As $\tau_{\text{energy}}$ increases, *Spe.* improves steadily, while *Eff.* and *Gen.* decline beyond a threshold. A larger $\tau_{\text{energy}}$ removes more of the shared component, suppressing unintended changes. If too large, however, it can also remove directions needed for the target edit, weakening the edit itself. Similarly, $\lambda$ shows a comparable trend, but with smaller variation. As $\lambda$ increases, *Eff.* remains stable, while *Spe.* improves modestly at the cost of a slight drop in *Gen.*. This is because a larger $\lambda$ strengthens the penalty $\lambda(\hat{\mathbf{w}}_1^\top \hat{\mathbf{w}}_2)^2$, encouraging $\mathbf{w}_1$ and $\mathbf{w}_2$ to be more distinct, which yields a more specific update. We extend this analysis to GPT-J and Qwen2.5, focusing on $\tau_{\text{energy}}$ (Appendix C.3). Both show trends similar to LLaMA3, but with lower sensitivity. Accordingly, we select hyperparameters by maximizing the harmonic mean score $S$ over *Eff.*, *Gen.*, and *Spe.*: $\lambda = 0.3$ for all models, and $\tau_{\text{energy}} = 0.4, 0.6$, and $0.6$ for LLaMA3, GPT-J, and Qwen2.5, respectively.

**Ablation Study.** We ablate the $\mathbf{k}'$ computation (§ 4.2) and the $\boldsymbol{\delta}'$ computation (§ 4.3) to assess their individual impacts (Table 6). While each component alone surpasses the strong baseline AlphaEdit in the harmonic mean score $S$, combining them in SUIT yields the strongest overall improvement, indicating that both components contribute meaningfully to the final performance and complement each other.

Table 6: $\mathbf{k}'$ and $\boldsymbol{\delta}'$ ablation results.

| Method | $S$ | *Eff.* | *Gen.* | *Spe.* |
|---|---|---|---|---|
| AlphaEdit | 55.8 | 97.3 | 88.7 | 31.0 |
| $\mathbf{k}'$ Only | 82.2 | 96.4 | 77.9 | 74.7 |
| $\boldsymbol{\delta}'$ Only | 68.3 | 99.7 | 83.8 | 44.6 |
| SUIT | 86.8 | 99.7 | 90.3 | 74.2 |

## 7 CONCLUSION

In this work, we introduced **Subspace Knowledge Edit (SUIT)**, a knowledge editing method that localizes target knowledge to edit-relevant subspaces and confines edits within them, improving specificity without sacrificing editing performance. Motivated by the *Linear Representation Hypothesis*, SUIT decomposes the key vector into entity-specific features and restricts the residual update vector $\boldsymbol{\delta}$ to features aligned with the target attribute. Across multiple models and evaluation settings, SUIT consistently improves *Specificity* and preserves overall model capabilities, while maintaining strong editing performance. These results suggest that explicitly modeling subspace structure is a promising direction for more precise and robust model editing.

ACKNOWLEDGMENTS

This work was supported by the National Research Foundation of Korea (NRF) under the grant RS-2024-00333484 and by the Institute of Information & Communications Technology Planning & Evaluation (IITP) under the grant RS-2024-00338140 (Development of Learning and Utilization Technology to Reflect Sustainability of Generative Language Models and Up-to-dateness over Time), all funded by the Korean government (MSIT). This work was also supported by the National Supercomputing Center with supercomputing resources including technical support (KSC-2025-CRE-0332, KSC-2025-CRE-0514).

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

## A  DETAILS OF THE ALPHAEDIT UPDATE FORMULA

### A.1  CLOSED-FORM SOLUTION

The primary objective is to find $\mathbf{\Delta}$ that incorporates new knowledge while preserving both the original model's knowledge and knowledge from previous edits. The optimization problem is formulated as follows:

$$\mathbf{\Delta} = \arg\min_{\tilde{\mathbf{\Delta}}} \left( \|\tilde{\mathbf{\Delta}}\mathbf{PK} - \mathbf{R}\|^2 + \|\tilde{\mathbf{\Delta}}\mathbf{P}\|^2 + \|\tilde{\mathbf{\Delta}}\mathbf{PK}_p\|^2 \right)$$

where the three terms correspond to the insertion of new information, a regularization term for stable convergence, and the preservation of prior edits, respectively.

This objective has a closed-form solution. The final update matrix $\mathbf{\Delta}' = \mathbf{\Delta P}$ is given by:

$$\mathbf{\Delta}' = \mathbf{RK}^\top \mathbf{P} \left( \mathbf{K}_p \mathbf{K}_p^\top \mathbf{P} + \mathbf{KK}^\top \mathbf{P} + \mathbf{I} \right)^{-1}$$

## A.2 Computation of the Projection Matrix $\mathbf{P}$

The matrix $\mathbf{P}$ is a projection matrix designed to project the update $\boldsymbol{\Delta}$ into the null space of a large key matrix $\mathbf{K}_0$, which represents a vast collection of the model's existing knowledge. This ensures that the update does not interfere with this preserved knowledge, satisfying $\boldsymbol{\Delta}\mathbf{P}\mathbf{K}_0 = \mathbf{0}$.

Due to the high dimensionality of $\mathbf{K}_0$, the projection is computed using the much smaller covariance matrix $\mathbf{K}_0\mathbf{K}_0^\top$. The procedure is as follows. First, Singular Value Decomposition (SVD) is performed on the covariance matrix:

$$\{\mathbf{U}, \boldsymbol{\Lambda}, (\mathbf{U})^\top\} = \text{SVD}(\mathbf{K}_0\mathbf{K}_0^\top)$$

Next, the eigenvectors in $\mathbf{U}$ (which are its columns) corresponding to near-zero eigenvalues are identified. A submatrix $\tilde{\mathbf{U}}$ is then constructed using only these selected eigenvectors. Finally, the projection matrix $\mathbf{P}$ is defined as:

$$\mathbf{P} = \tilde{\mathbf{U}}(\tilde{\mathbf{U}})^\top$$

## A.3 Computation of the Prior Keys Matrix $\mathbf{K}_p$

The matrix $\mathbf{K}_p$ is used in sequential editing tasks to protect the knowledge updated in previous steps from being disrupted by the current edit. It is constructed by aggregating the key matrices from all prior edits.

Specifically, if there have been $t - 1$ previous edits, with corresponding key matrices $\mathbf{K}_1, \mathbf{K}_2, \ldots, \mathbf{K}_{t-1}$, then $\mathbf{K}_p$ is the horizontal concatenation of these matrices:

$$\mathbf{K}_p = [\mathbf{K}_1, \mathbf{K}_2, \ldots, \mathbf{K}_{t-1}]$$

For the very first edit, $\mathbf{K}_p$ is an empty matrix.

## A.4 The Regularization Term $\mathcal{R}$

The optimization objective to find the residual vector $\boldsymbol{\delta}$ is given by:

$$\boldsymbol{\delta} = \arg\min_{\tilde{\boldsymbol{\delta}}} \left\{ -\log p\left(a^* \mid \mathbf{h}^* \leftarrow \mathbf{h} + \tilde{\boldsymbol{\delta}}\right) + \mathcal{R} \right\}.$$

The regularization term $\mathcal{R}$ is introduced to prevent the model from overfitting to the new fact, which could corrupt existing knowledge. It consists of two components: a KL divergence term and a weight decay term. The full regularization term is formulated as:

$$\mathcal{R} = \lambda_{\text{KL}} D_{\text{KL}}(p(\mathbf{h}) \,\|\, p(\mathbf{h} + \tilde{\boldsymbol{\delta}})) + \lambda_{\text{WD}}\|\tilde{\boldsymbol{\delta}}\|_2^2.$$

**KL Divergence.** The first term uses the Kullback-Leibler (KL) divergence to preserve knowledge related to the entity of the edit. This is achieved by computing the divergence on a prompt, such as *"{entity} is a"* (e.g., *"Chrome is a"*). Specifically, it measures the divergence between the output probability distribution from the original hidden state $\mathbf{h}$ and the distribution from the modified hidden state $\mathbf{h} + \tilde{\boldsymbol{\delta}}$. The hyperparameter $\lambda_{\text{KL}}$ controls the strength of this penalty.

**Weight Decay.** The second term is a weight decay penalty on the L2 norm of the residual vector $\tilde{\boldsymbol{\delta}}$. This term encourages the model to find a smaller solution for $\tilde{\boldsymbol{\delta}}$. The hyperparameter $\lambda_{\text{WD}}$ controls the strength of this penalty.

# B The Additive Update

The updated residual stream $\mathbf{h}^*$ is computed via a simple additive update:

$$\mathbf{h}^* = \mathbf{h} + \boldsymbol{\delta}',$$

where our residual vector $\boldsymbol{\delta}'$ is defined as:

$$\boldsymbol{\delta}' = (\mathbf{h}^\top\mathbf{w}_2 - \mathbf{h}^\top\mathbf{w}_1)\mathbf{w}_1 + (\mathbf{h}^\top\mathbf{w}_1 - \mathbf{h}^\top\mathbf{w}_2)\mathbf{w}_2.$$

This formulation is simplified by ignoring interactions between the feature directions $\mathbf{w}_1$ and $\mathbf{w}_2$. By assuming interactions between them to be zero (i.e., $\mathbf{w}_1^\top \mathbf{w}_2 = 0$), we can achieve the intended swap of magnitudes with a straightforward additive operation.

To verify that this update swaps the magnitudes of $\mathbf{h}$ along $\mathbf{w}_1$ and $\mathbf{w}_2$, we can compute the new projections $(\mathbf{h}^*)^\top \mathbf{w}_1$ and $(\mathbf{h}^*)^\top \mathbf{w}_2$. Since $\mathbf{w}_1$ and $\mathbf{w}_2$ are unit vectors, $\mathbf{w}_1^\top \mathbf{w}_1 = 1$ and $\mathbf{w}_2^\top \mathbf{w}_2 = 1$.

First, let's compute the projection of $\mathbf{h}^*$ onto $\mathbf{w}_1$:

$$
\begin{aligned}
(\mathbf{h}^*)^\top \mathbf{w}_1 &= \left(\mathbf{h} + (\mathbf{h}^\top \mathbf{w}_2 - \mathbf{h}^\top \mathbf{w}_1)\mathbf{w}_1 + (\mathbf{h}^\top \mathbf{w}_1 - \mathbf{h}^\top \mathbf{w}_2)\mathbf{w}_2\right)^\top \mathbf{w}_1 \\
&= \mathbf{h}^\top \mathbf{w}_1 + (\mathbf{h}^\top \mathbf{w}_2 - \mathbf{h}^\top \mathbf{w}_1)(\mathbf{w}_1^\top \mathbf{w}_1) + (\mathbf{h}^\top \mathbf{w}_1 - \mathbf{h}^\top \mathbf{w}_2)(\mathbf{w}_2^\top \mathbf{w}_1) \\
&= \mathbf{h}^\top \mathbf{w}_1 + (\mathbf{h}^\top \mathbf{w}_2 - \mathbf{h}^\top \mathbf{w}_1)(1) + (\mathbf{h}^\top \mathbf{w}_1 - \mathbf{h}^\top \mathbf{w}_2)(0) \\
&= \mathbf{h}^\top \mathbf{w}_1 + \mathbf{h}^\top \mathbf{w}_2 - \mathbf{h}^\top \mathbf{w}_1 \\
&= \mathbf{h}^\top \mathbf{w}_2
\end{aligned}
$$

As shown, the new magnitude of the hidden state along $\mathbf{w}_1$ is equal to its original magnitude along $\mathbf{w}_2$.

Next, we compute the projection of $\mathbf{h}^*$ onto $\mathbf{w}_2$:

$$
\begin{aligned}
(\mathbf{h}^*)^\top \mathbf{w}_2 &= \left(\mathbf{h} + (\mathbf{h}^\top \mathbf{w}_2 - \mathbf{h}^\top \mathbf{w}_1)\mathbf{w}_1 + (\mathbf{h}^\top \mathbf{w}_1 - \mathbf{h}^\top \mathbf{w}_2)\mathbf{w}_2\right)^\top \mathbf{w}_2 \\
&= \mathbf{h}^\top \mathbf{w}_2 + (\mathbf{h}^\top \mathbf{w}_2 - \mathbf{h}^\top \mathbf{w}_1)(\mathbf{w}_1^\top \mathbf{w}_2) + (\mathbf{h}^\top \mathbf{w}_1 - \mathbf{h}^\top \mathbf{w}_2)(\mathbf{w}_2^\top \mathbf{w}_2) \\
&= \mathbf{h}^\top \mathbf{w}_2 + (\mathbf{h}^\top \mathbf{w}_2 - \mathbf{h}^\top \mathbf{w}_1)(0) + (\mathbf{h}^\top \mathbf{w}_1 - \mathbf{h}^\top \mathbf{w}_2)(1) \\
&= \mathbf{h}^\top \mathbf{w}_2 + \mathbf{h}^\top \mathbf{w}_1 - \mathbf{h}^\top \mathbf{w}_2 \\
&= \mathbf{h}^\top \mathbf{w}_1
\end{aligned}
$$

Similarly, the new magnitude along $\mathbf{w}_2$ becomes the original magnitude along $\mathbf{w}_1$. Thus, this simple additive update, which ignores interactions between the two directions, effectively swaps the magnitudes as intended.

## C  EXPERIMENTAL DETAILS

### C.1  BASELINE METHODS

Below we provide brief descriptions of the baseline methods used for comparison. Our main set comprises Fine-Tuning (FT; FT-L/FT-W) (Zhu et al., 2020), MEND (Mitchell et al., 2022a), PMET (Li et al., 2024), MEMIT (Meng et al., 2023b), and AlphaEdit (Fang et al., 2025). Additional baselines include ROME (Meng et al., 2023a), RECT (Gu et al., 2024), PRUNE (Ma et al., 2024), and NSE (Jiang et al., 2024).

**Fine-Tuning (FT-L & FT-W).**  *FT-L* fine-tunes only the weights of a specific layer (as identified by ROME), rather than all layers. *FT-W* is a variant of FT-L that differs slightly in the loss used for parameter optimization under regularization.

**MEND (Model Editor Networks with Gradient Decomposition).**  Edits large pre-trained models from a single input–output pair by applying a low-rank decomposition to the fine-tuning gradient and using small auxiliary "editor" networks for fast, localized parameter updates that mitigate overfitting.

**PMET (Precise Model Editing in Transformers).**  Observes that hidden states arise from FFN, MHSA, and residual paths. It assumes MHSA encodes general extraction patterns and need not be altered; PMET jointly optimizes hidden states for FFN/MHSA but updates only FFN weights using the optimized FFN state to make more precise edits.

**MEMIT (Mass-Editing Memory in a Transformer).**  Extends ROME to insert many new factual memories efficiently by targeting transformer modules that causally mediate factual recall, enabling simultaneous updates for thousands of associations.

**AlphaEdit.** Within the locate–then–edit paradigm, projects the parameter perturbation onto the null space of knowledge to be preserved before applying it, so outputs for preserved queries remain unchanged and corruption during sequential edits is reduced.

**ROME (Rank-One Model Editing).** Identifies key mid-layer feed-forward activations that influence factual predictions and applies a direct rank-one weight update to modify specific factual associations.

**RECT (Regularizing Causal Tracing).** Regularizes weight updates during editing to prevent excessive changes and overfitting, mitigating side effects (e.g., reasoning degradation) while maintaining general capabilities.

**PRUNE (Preserving Representations through Unitary Nullspace Editing).** Constrains the edited matrix (e.g., via condition-number control and null-space restrictions) so perturbations remain limited to stored knowledge, preserving overall ability under sequential edits.

**NSE (Neuron-level Sequential Editing).** It improves sequential editing by computing target values with original weights for value computation and by updating only activation-based influential neurons at the neuron level, rather than modifying the full weight matrix.

### C.2 Hyperparameter Settings

We use a unified hyperparameter configuration across *locate–then–edit* methods (ROME, MEMIT, PMET, RECT, PRUNE, NSE), as well as AlphaEdit and SUIT, and deviate only when method-specific constraints apply. Following prior *locate–then–edit* work (Meng et al., 2023a;b), we set the two hyperparameters described in Appendix A.4, $\lambda_{\text{KL}}$ and $\lambda_{\text{WD}}$, to 0.0625 and 0.5, respectively. SUIT does not use $\lambda_{\text{KL}}$ or $\lambda_{\text{WD}}$, and therefore does not require these hyperparameters.

For model-specific layer selection, following prior work (Meng et al., 2023a;b), we use zero-based layer indices $\{4, 5, 6, 7, 8\}$ for LLaMA3-Instruct (8B), $\{3, 4, 5, 6, 7, 8\}$ for GPT-J (6B), and $\{4, 5, 6, 7, 8\}$ for Qwen2.5-Instruct (7B). Consistent with the original ROME setup, ROME edits only a single target layer (index 5). In addition, following AlphaEdit, we set the null-space threshold to 0.02 and the L2 regularization coefficient to 10 for both AlphaEdit and SUIT. In AlphaEdit, the null-space threshold controls the strength of the preservation constraint, with smaller values imposing a tighter constraint.

For SUIT, as described in §6.3, we sweep the hyperparameters over $\{0.1, 0.2, \ldots, 0.9\}$ and select the values that maximize the harmonic mean score $S$, which balances *Eff.*, *Gen.*, and *Spe.*. Specifically, we use $\lambda = 0.3$ for all models, and set $\tau_{\text{energy}} = 0.4$ for LLaMA3 and $\tau_{\text{energy}} = 0.6$ for GPT-J and Qwen2.5.

For methods outside the *locate–then–edit* family, MEND and FT-L/FT-W are tuned following their original papers. Finally, for hyperparameters not explicitly specified in this section, we follow the original works exactly. When such settings are not provided in the original papers or code, we default to the settings used in the EASYEDIT open-source framework, which provides a unified implementation of knowledge editing methods Wang et al. (2024a).

### C.3 $\tau_{\text{ENERGY}}$ Hyperparameter Analysis on GPT-J and Qwen2.5

In Fig. 5, our analysis on LLaMA3 shows that $\tau_{\text{energy}}$ has a sensitive effect on editing metrics. Motivated by this observation, we conduct the same hyperparameter analysis on GPT-J and Qwen2.5 in this section. To compare sensitivity to $\tau_{\text{energy}}$, we follow the same setup as in Fig. 5: evaluation on COUNTERFACT with 1,000 edits (batch size 100), while fixing $\lambda = 0.3$ and varying $\tau_{\text{energy}}$. Both models exhibit the same qualitative trend as LLaMA3: as $\tau_{\text{energy}}$ increases, *Spe.* generally improves, while excessively large values can reduce *Eff.* and *Gen.*. However, for GPT-J and Qwen2.5, the trend is less pronounced than for LLaMA3, indicating relatively lower sensitivity to $\tau_{\text{energy}}$. Table 7 reports the full sweep results. Accordingly, we select hyperparameters using the same criterion as in the main text, namely by maximizing the harmonic mean score $S$ over *Eff.*, *Gen.*, and *Spe.*. As shown in Table 7, $\tau_{\text{energy}} = 0.6$ achieves the highest $S$ for both GPT-J and Qwen2.5, and we use this value for both models.

Table 7: $\tau_{\text{energy}}$ sweep on GPT-J and Qwen2.5.

| $\tau_{\text{energy}}$ | GPT-J | | | | Qwen2.5 | | | |
|---|---|---|---|---|---|---|---|---|
| | $S$ | Eff. | Gen. | Spe. | $S$ | Eff. | Gen. | Spe. |
| 0.1 | 82.54 | 99.10 | 94.70 | 63.71 | 85.42 | 99.30 | 91.05 | 71.08 |
| 0.2 | 82.71 | 99.10 | 94.70 | 64.01 | 85.26 | 99.30 | 91.20 | 70.66 |
| 0.3 | 83.11 | 99.10 | 94.80 | 64.69 | 84.99 | 99.30 | 90.70 | 70.41 |
| 0.4 | 84.00 | 99.10 | 94.90 | 66.28 | 85.26 | 99.20 | 90.35 | 71.24 |
| 0.5 | 84.23 | 99.10 | 94.70 | 66.82 | 85.40 | 99.30 | 91.05 | 71.04 |
| 0.6 | **84.67** | 99.20 | 94.60 | 67.66 | **86.12** | 99.20 | 91.55 | 72.30 |
| 0.7 | 84.54 | 99.10 | 94.25 | 67.63 | 85.88 | 99.30 | 91.00 | 72.08 |
| 0.8 | 84.61 | 99.10 | 93.45 | 68.19 | 85.45 | 99.10 | 91.00 | 71.29 |
| 0.9 | 84.64 | 99.10 | 93.05 | 68.46 | 86.10 | 99.20 | 91.00 | 72.61 |

## D   DETAILED DESCRIPTION OF EVALUATION METRICS

Let $a$ denote the old attribute and $a^*$ the new attribute. For each item $i$, let $x_i$ be the rewrite prompt (i.e., $(e_i, r_i)$), $N(x_i)$ its paraphrase prompts, and $O(x_i)$ its neighborhood prompts. All probabilities $P_{f_\theta}(\cdot \mid \cdot)$ are computed under the language model $f_\theta$. These evaluation metrics are not new; we follow established practice from prior work (Fang et al., 2025; Meng et al., 2023a;b).

### D.1   COUNTERFACT METRICS

**Probability-based criterion.**   The following three metrics use probability comparisons between the edited target $a^*$ and the original $a$.

**Efficacy.**

$$\mathbb{E}_i \, \mathbf{1}\Big[P_{f_\theta}\big(a^* \mid x_i\big) \; > \; P_{f_\theta}\big(a \mid x_i\big)\Big].$$

**Generalization.**

$$\mathbb{E}_i \, \mathbf{1}\Big[P_{f_\theta}\big(a^* \mid N(x_i)\big) \; > \; P_{f_\theta}\big(a \mid N(x_i)\big)\Big].$$

**Specificity.**

$$\mathbb{E}_i \, \mathbf{1}\Big[P_{f_\theta}\big(a \mid O(x_i)\big) \; > \; P_{f_\theta}\big(a^* \mid O(x_i)\big)\Big].$$

**Generation-based criterion (exact match).**   Let $\tau(a^*) = (a_1^*, \ldots, a_{T^*}^*)$. Success if every target token is the greedy choice at its step:

$$\mathbb{E}_i \, \mathbf{1}\left[\forall t \in \{1, \ldots, T^*\} : \; a_t^* \; = \; \arg\max_y \, P_{f_\theta}\big(y \mid a_{<t}^*, \, x_i\big)\right].$$

**Fluency (generation entropy).**   A measure for excessive repetition in model outputs. It uses the entropy of n-gram distributions:

$$-\frac{2}{3} \sum_k g_2(k) \, \log_2 g_2(k) \; + \; \frac{4}{3} \sum_k g_3(k) \, \log_2 g_3(k), \tag{22}$$

where $g_n(\cdot)$ is the n-gram frequency distribution.

**Consistency (reference score).**   The consistency of the model's outputs is evaluated by giving the model $f_\theta$ an entity $e$ and computing the cosine similarity between the TF-IDF vectors of the model-generated text and a reference Wikipedia text about $a$.

We report the results based on the generation-based criterion in Table 1, as it serves as a more stringent evaluation and more directly reflects deployment-relevant behavior: in practice, edited knowledge is only actionable when it is realized in the model's generated responses. For direct comparison with the traditional probability-based metrics commonly used in prior literature, we additionally report those results in Table 9.

## D.2 ZSRE METRICS

**Token-level partial credit.** For target string $y$ with tokenization $\tau(y) = (y_1, \ldots, y_{|y|})$ and prompt $x$, define the token-level accuracy under teacher-forced greedy decoding as

$$\text{TokenAcc}(x, y) \;=\; \frac{1}{|y|} \sum_{t=1}^{|y|} \mathbf{1}\Big[\, y_t \;=\; \arg\max_v \, P_{f_\theta}\big(v \,\big|\, y_{<t}, \, x\big) \Big].$$

**Efficacy.** Average token-level accuracy on rewrite prompts:

$$\mathbb{E}_i\big[\,\text{TokenAcc}\big(x_i, \, a^*\big)\,\big].$$

**Generalization.** Average token-level accuracy on paraphrase prompts:

$$\mathbb{E}_i\big[\,\text{TokenAcc}\big(N(x_i), \, a^*\big)\,\big].$$

**Specificity.** For zsRE, we evaluate *Specificity* using exact match against the *pre-edit model*'s output on neighborhood prompts, rather than the dataset gold answer. This is because, as discussed in Appendix D.3, models often fail to exactly reproduce the zsRE gold answer even before editing, making gold-referenced exact match a weak preservation signal.

Let $a_i^{\text{pre}}$ be the output generated by the pre-edit model for $O(x_i)$, and let $a_i^{\text{post}}$ be the output generated by the post-edit model for the same prompt. Then, *Specificity* is defined as:

$$\mathbb{E}_i \, \mathbf{1}\Big[ a_i^{\text{post}} \;=\; a_i^{\text{pre}} \Big].$$

## D.3 ZSRE DATASET DETAILS AND RATIONALE FOR OUR EVALUATION PROTOCOL

In this section, we explain why we cannot apply the same pre-knowledge filtering used for COUNTERFACT to ZSRE. This issue arises from the question-style prompts and response format of ZSRE, and therefore requires a separate evaluation protocol.

As described in § 5.1, for COUNTERFACT we evaluate only instances where the pre-edit model generates the ground-truth answer. This ensures that *Efficacy*/*Generalization* reflect edits to known facts (not insertion), while *Specificity* measures preservation of previously exhibited knowledge. However, unlike declarative datasets such as COUNTERFACT, ZSRE uses question-style prompts, making the same pre-knowledge filtering unreliable. Without additional instructions, the model often does not directly generate the dataset ground-truth answer span in a short attribute-only form. For example, for the prompt *"What airport is China United Airlines part of?"*, the gold attribute is *"Beijing Nanyuan Airport,"* but the model may instead generate a longer sentence such as *"China United Airlines is part of Beijing Capital International Airport."*

To examine this mismatch more concretely, we analyze the first-token distribution. For LLaMA3, zsRE answers frequently begin with function words or pronouns unrelated to the gold attribute (e.g., "the" 4,902 times, "a" 445 times, "he" 388 times, and "she" 89 times). This makes an argmax-based pre-knowledge check unreliable, since the earliest generated tokens often do not correspond to the target attribute span. As a result, ground-truth-generation-based pre-knowledge filtering leaves almost no valid instances on ZSRE; for LLaMA3, only 144 of 19,086 instances remain when we additionally require ground-truth generation for the neighborhood prompts. Accordingly, we evaluate ZSRE without this pre-knowledge condition. Consequently, because the dataset ground-truth answer does not necessarily reflect knowledge actually exhibited by the pre-edit model on ZSRE, we do not use it as the preservation reference for *Specificity*. Instead, as described in Appendix D.2, we use the model's pre-edit generated response to the prompt as the reference and evaluate *Specificity* by exact match.

# E ADDITIONAL EXPERIMENTAL RESULTS

## E.1 EXTENDED BASELINE COMPARISONS

Table 8 presents results for additional extended baselines that were not reported in Table 1.

Table 8: Results on COUNTERFACT (batch size = 100).

| Model | Method | COUNTERFACT | | | | | | |
|---|---|---|---|---|---|---|---|---|
| | | S ↑ | Eff. ↑ | Gen. ↑ | Spe. ↑ | Flu. ↑ | Con. ↑ | GC ↑ |
| LLaMA3 | Pre-edit | 0.0 | 0.0 | 0.0 | 100.0 | 634.9 | 20.9 | 62.3 |
| | FT-W | 4.9 | 4.0 | 2.9 | 43.5 | **634.4** | 21.4 | 60.5 |
| | ROME | 0.0 | 0.0 | 0.2 | 0.0 | 481.5 | 4.2 | 0.0 |
| | RECT | 55.8 | 81.6 | 72.4 | 36.1 | **634.4** | 35.3 | 60.4 |
| | PRUNE | 28.6 | 43.3 | 39.3 | 17.7 | 590.4 | 33.9 | 45.0 |
| | NSE | 3.3 | 1.4 | 5.8 | 62.2 | 609.6 | 23.1 | 60.9 |
| | SUIT | **86.8** | **99.7** | **90.3** | **74.2** | 631.2 | **38.2** | **61.8** |
| GPT-J | Pre-edit | 0.0 | 0.0 | 0.0 | 100.0 | 621.1 | 23.9 | 18.6 |
| | FT-W | 5.8 | 12.3 | 2.4 | 49.0 | 613.4 | 25.7 | **36.1** |
| | ROME | 0.2 | 0.1 | 0.2 | 0.2 | 407.2 | 4.2 | 0.0 |
| | RECT | 66.8 | 92.9 | 85.8 | 44.4 | **625.0** | 47.6 | 14.9 |
| | PRUNE | 31.1 | 51.8 | 56.0 | 16.9 | 504.6 | 29.4 | 29.4 |
| | NSE | 2.2 | 0.8 | 12.6 | 54.1 | 608.0 | 34.2 | 27.5 |
| | SUIT | **82.3** | **98.6** | **93.3** | **64.1** | 619.4 | **49.4** | 17.8 |
| Qwen2.5 | Pre-edit | 0.0 | 0.0 | 0.0 | 100.0 | 625.5 | 21.9 | 20.8 |
| | FT-W | 10.3 | 47.9 | 31.7 | 4.2 | 476.7 | 4.5 | 0.0 |
| | ROME | 25.1 | 51.6 | 33.7 | 14.2 | 440.1 | 15.6 | 0.6 |
| | RECT | 64.0 | 86.3 | 85.9 | 42.3 | 625.8 | **37.7** | **59.9** |
| | PRUNE | 15.2 | 28.2 | 30.7 | 7.7 | 588.1 | 30.4 | 6.0 |
| | NSE | 0.0 | 0.0 | 0.0 | **99.5** | 625.6 | 21.7 | 39.3 |
| | SUIT | **85.7** | **99.5** | **86.8** | 74.4 | **626.2** | 37.4 | 23.7 |

Table 9: Probability-based criterion results (Eff./Gen./Spe.) on three models. *S* denotes the harmonic mean of *Eff.*, *Gen.*, and *Spe.*

| LLaMA3 (Prob) | | | | |
|---|---|---|---|---|
| Method | S ↑ | Eff. ↑ | Gen. ↑ | Spe. ↑ |
| Pre-Edit | 0.0 | 0.0 | 0.0 | 100.0 |
| FT-W | 11.3 | 9.5 | 6.7 | 92.4 |
| ROME | 59.5 | 67.1 | 65.5 | 49.4 |
| RECT | 84.2 | 94.0 | 87.2 | 73.9 |
| PRUNE | 69.5 | 76.5 | 75.1 | 59.7 |
| NSE | 73.2 | 83.7 | 53.1 | **98.0** |
| FT-L | 60.1 | 93.0 | 89.1 | 35.8 |
| MEND | 51.3 | 52.7 | 53.1 | 48.4 |
| MEMIT | 82.6 | 90.8 | 88.8 | 71.1 |
| PMET | 76.2 | 82.7 | 81.0 | 67.0 |
| AlphaEdit | 87.2 | 99.7 | **94.1** | 72.7 |
| SUIT | **91.3** | **100.0** | 90.8 | 84.4 |

| GPT-J (Prob) | | | | |
|---|---|---|---|---|
| Method | S ↑ | Eff. ↑ | Gen. ↑ | Spe. ↑ |
| Pre-Edit | 0.0 | 0.0 | 0.0 | 100.0 |
| FT-W | 13.6 | 21.8 | 6.1 | 88.4 |
| ROME | 51.2 | 49.6 | 48.9 | 55.6 |
| RECT | 87.1 | 98.5 | 92.8 | 74.0 |
| PRUNE | 72.6 | 87.2 | 88.3 | 53.9 |
| NSE | 82.3 | 88.5 | 70.6 | **90.9** |
| FT-L | 61.8 | 91.1 | 78.3 | 40.3 |
| MEND | 48.4 | 46.0 | 46.1 | 53.9 |
| MEMIT | 85.0 | 97.9 | 96.8 | 67.8 |
| PMET | 83.5 | 93.7 | 94.5 | 68.2 |
| AlphaEdit | 91.0 | **99.6** | **97.9** | 78.6 |
| SUIT | **94.9** | 99.3 | 96.2 | 89.7 |

| Qwen2.5 (Prob) | | | | |
|---|---|---|---|---|
| Method | S ↑ | Eff. ↑ | Gen. ↑ | Spe. ↑ |
| Pre-Edit | 0.0 | 0.0 | 0.0 | 100.0 |
| FT-W | 54.7 | 82.3 | 71.7 | 34.8 |
| ROME | 66.5 | 76.7 | 69.2 | 56.8 |
| RECT | 65.6 | 95.9 | 85.9 | 42.3 |
| PRUNE | 66.6 | 72.2 | 73.6 | 56.7 |
| NSE | 0.0 | 0.0 | 0.0 | **100.0** |
| FT-L | 54.7 | 82.3 | 71.7 | 34.8 |
| MEND | 51.0 | 54.0 | 53.9 | 46.0 |
| MEMIT | 86.9 | 92.2 | 95.4 | 75.8 |
| PMET | 76.4 | 85.8 | 86.9 | 62.1 |
| AlphaEdit | 91.7 | 99.2 | **98.3** | 80.3 |
| SUIT | **96.4** | **100.0** | 94.4 | 95.0 |

## E.2 DETAILED F1 SCORES ON GENERAL CAPABILITY BENCHMARKS

### E.2.1 GENERAL CAPABILITY BENCHMARK DATASETS

To evaluate the general capabilities of language models, several well-known benchmark datasets are utilized. The **GLUE** (General Language Understanding Evaluation) benchmark is a prominent collection of diverse natural language understanding tasks (Wang et al., 2019). Key datasets included in GLUE are: **SST-2** (The Stanford Sentiment Treebank), a single-sentence classification task for sentiment analysis of movie reviews (Socher et al., 2013); **MRPC** (Microsoft Research Paraphrase Corpus), which involves determining if a pair of sentences are semantically equivalent (Dolan & Brockett, 2005); **RTE** (Recognizing Textual Entailment), a task that assesses whether a premise sentence logically entails a hypothesis (Bentivogli et al., 2009); and **CoLA** (Corpus of Linguistic Acceptability), where the task is to decide if a sentence is grammatically acceptable (Warstadt et al., 2019). Furthermore, **NLI** (Natural Language Inference) tasks, which require inferring the logical relationship (entailment, contradiction, or neutral) between a pair of sentences, are a crucial part of the evaluation (Williams et al., 2018).

Beyond GLUE, more comprehensive benchmarks exist to measure multi-task proficiency. **MMLU** (Massive Multi-task Language Understanding) is a benchmark designed to measure a text model's multi-task accuracy under zero-shot and few-shot settings across a wide range of subjects (Hendrycks et al., 2021).

### E.2.2 General Capability Benchmark Results

Table 10 reports the F1 scores for each benchmark in the General Capability evaluation.

Table 10: F1 scores per benchmark.

| Model | Method | SST | MMLU | MRPC | CoLA | RTE | NLI | Avg. |
|---|---|---|---|---|---|---|---|---|
| LLaMA3 | Pre-Edit | 81.78 | 59.93 | 65.28 | 76.72 | 27.65 | 69.27 | 63.44 |
| | FT-L | 0.00 | 0.00 | 37.34 | 0.00 | 0.00 | 0.00 | 6.22 |
| | MEND | 0.00 | 0.00 | 0.00 | 0.00 | 0.00 | 0.00 | 0.00 |
| | MEMIT | 75.28 | 55.93 | 64.80 | 69.57 | 31.46 | 67.90 | 60.82 |
| | PMET | 64.25 | 28.33 | 60.70 | 55.52 | **32.16** | 59.55 | 50.09 |
| | AlphaEdit | **77.87** | 57.82 | 61.72 | 76.36 | 31.52 | 67.64 | 62.16 |
| | SUIT | 77.18 | **58.93** | **65.64** | **77.63** | 29.53 | **69.26** | **63.03** |
| GPT-J | Pre-Edit | 0.00 | 5.78 | 23.16 | 21.41 | 42.67 | 52.78 | 24.30 |
| | FT-L | 0.00 | **17.74** | 19.67 | 15.92 | **46.13** | 45.79 | **24.21** |
| | MEND | 0.00 | 0.00 | 0.00 | 0.00 | 0.00 | 0.00 | 0.00 |
| | MEMIT | 0.00 | 5.89 | 6.73 | 13.29 | 41.56 | **53.33** | 20.13 |
| | PMET | 0.00 | 7.81 | 3.63 | 12.22 | 44.69 | 46.17 | 19.09 |
| | AlphaEdit | 0.00 | 4.30 | 5.81 | 9.26 | 44.91 | 52.79 | 19.51 |
| | SUIT | 0.00 | 9.44 | **24.20** | **21.78** | 33.66 | 33.03 | 20.35 |
| Qwen2.5 | Pre-Edit | 13.66 | 2.46 | 53.32 | 23.36 | 11.21 | 69.23 | 28.87 |
| | FT-L | 0.00 | 0.00 | 0.00 | 0.00 | 0.00 | 0.00 | 0.00 |
| | MEND | 0.00 | 0.00 | 0.00 | 0.00 | 0.00 | 0.00 | 0.00 |
| | MEMIT | 54.31 | 0.79 | **55.82** | 24.85 | 11.49 | 61.48 | **34.79** |
| | PMET | 15.15 | **14.78** | 2.27 | 3.07 | **27.16** | 26.50 | 14.82 |
| | AlphaEdit | **59.25** | 0.39 | 39.63 | 6.47 | 11.56 | 51.52 | 28.14 |
| | SUIT | 19.07 | 3.23 | 55.03 | **32.27** | 8.80 | **66.23** | 30.77 |

### E.3 Detailed Results for the 5,000-Edit Setting on CounterFact

Fig. 1 reports the average performance across all benchmarks, while Fig. 6 presents the F1 scores for each benchmark individually. Table 11 presents the performance metrics on the CounterFact dataset, where 5,000 cases were sequentially edited in batches of 100.

Table 11: Full performance metrics on COUNTERFACT dataset for LLaMA3. Metrics are grouped by a generation-based criterion and a probability-based criterion.

| **Method** | Generation-based | | | | Probability-based | | | | *Flu.* ↑ | *Con.* ↑ |
|---|---|---|---|---|---|---|---|---|---|---|
| | $S$ ↑ | *Eff.* ↑ | *Gen.* ↑ | *Spe.* ↑ | $S$ ↑ | *Eff.* ↑ | *Gen.* ↑ | *Spe.* ↑ | | |
| SUIT | **38.0** | **95.8** | 58.2 | **19.5** | **89.9** | **99.0** | 89.0 | **83.2** | **624.1** | 33.5 |
| AlphaEdit | 28.9 | 89.7 | **69.4** | 12.8 | 80.6 | 97.6 | **92.9** | 61.7 | 613.9 | **33.6** |

### E.4 Scaling to a Larger Model

We further evaluate SUIT on a larger-scale model to examine whether its benefits persist beyond the model sizes considered in Table 1. Specifically, we run additional experiments on LLaMA2-13B (Touvron et al., 2023) using COUNTERFACT. Since prior knowledge-editing baselines do not provide tuned hyperparameters or reported results at this scale, we ensure comparability by reusing exactly the same hyperparameter settings as those used for LLaMA3-8B (Appendix C.2) for all methods,

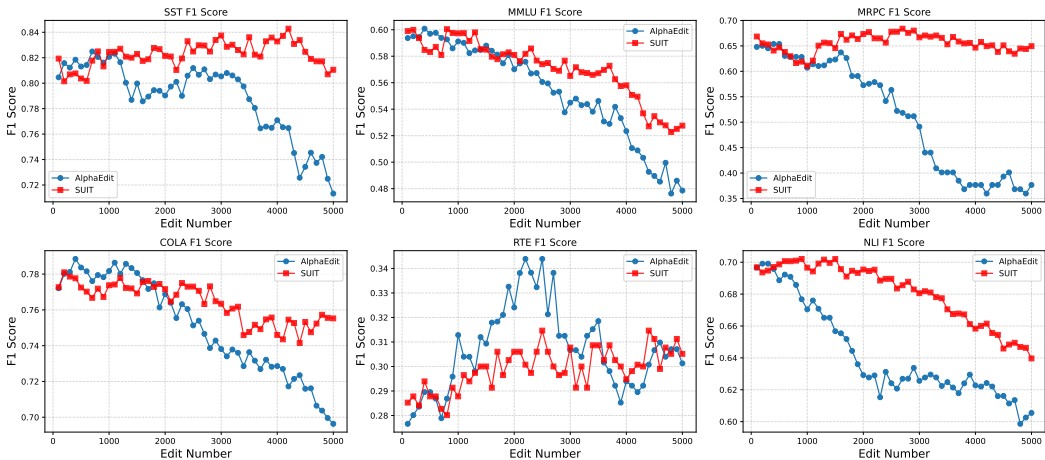

Figure 6: F1 scores for each benchmark.

including SUIT. We then evaluate every method under the sequential batch-editing setup (5 batches × 100 edits). Table 12 shows that SUIT achieves the highest $S$ score, with a substantial improvement in Specificity and a concurrent gain in Generalization. These results suggest that SUIT scales favorably to larger models while maintaining a strong balance among Efficacy, Generalization, and Specificity, even without any model-specific retuning.

Table 12: Results on LLaMA2-13B.

| Method | $S$ | *Eff.* | *Gen.* | *Spe.* |
|---|---|---|---|---|
| MEMIT | 73.2 | 90.0 | 68.3 | 65.6 |
| AlphaEdit | 79.3 | **98.4** | 76.5 | 68.4 |
| SUIT | **84.7** | 95.0 | **79.1** | **81.7** |

## E.5    EVALUATION ON CHED (CONTEXTUAL HOP EDITING DATASET)

Table 13: Performance comparison on CHED. Each column corresponds to a rewrite prompt augmented with one of six prefix–context types: *Subject*, *Obj-Old*, *Obj-New*, and their 1-hop variants.

| Context Types | Subject | Obj-Old | Obj-New | Subject Hop | Obj-Old Hop | Obj-New Hop | Avg. |
|---|---|---|---|---|---|---|---|
| MEMIT | 75.4 | 73.6 | 77.8 | 74.1 | 70.4 | 75.7 | 74.5 |
| AlphaEdit | 92.7 | 88.6 | 94.2 | 90.4 | 87.9 | **92.0** | 91.0 |
| SUIT | **95.7** | **92.0** | **95.6** | **94.3** | **91.2** | 93.4 | **93.7** |

CHED (Park et al., 2025) extends COUNTERFACT by evaluating whether knowledge edits remain robust under additional prefix contexts. Specifically, each rewrite prompt $(e, r)$ is preceded by sentences derived from either the original entity $e$, the old attribute $a$, the new attribute $a^*$, or their one-hop neighbors. The six context types thus test whether the edited model can maintain correctness when auxiliary but semantically related cues are introduced. As shown in Table 13, SUIT consistently outperforms across all context types, indicating strong resilience to contextual variation.

## E.6    RIPPLEEDITS EVALUATION RESULTS

We evaluate ripple effects using the RippleEdits benchmark (Cohen et al., 2024), which measures whether a single factual edit correctly propagates to logically or compositionally implied facts while preserving unrelated knowledge. We follow the benchmark's standard setup and test 100 single edits from the POPULAR split. This split consists of high-frequency entities and represents a higher-severity

Table 14: RippleEdits results on 100 single edits from the POPULAR split.

| Method | LG | CI | CII | SA | PV | RS |
|---|---|---|---|---|---|---|
| AlphaEdit | 0.210 | 0.482 | 0.000 | 1.000 | 1.000 | 0.597 |
| SUIT | 0.257 | 0.497 | 0.000 | 1.000 | 1.000 | 0.500 |
| MEMIT | 0.188 | 0.256 | 0.000 | 0.131 | 0.074 | 0.220 |

regime that is more likely to surface ripple errors. We report accuracy on the six RippleEdits criteria: Logical Generalization (LG), Compositionality I/II (CI/CII), Subject Aliasing (SA), Preservation (PV), and Relation Specificity (RS).

Overall, SUIT clearly outperforms MEMIT on this challenging split and remains broadly competitive with AlphaEdit, indicating strong robustness to ripple effects even in a high-severity setting.

## F  DETAILED VISUALIZATION OF COMPONENT EFFECTS

As stated in § 4.3 and § 6.2.2, we posited that $\mathbf{w}_1$ increases the logit of the new attribute $a^*$ and $\mathbf{w}_2$ decreases the logit of the original attribute $a$. To test this, we analyzed the individual roles of $\mathbf{w}_1$ and $\mathbf{w}_2$ by decomposing our residual vector $\boldsymbol{\delta}'$ into its components:

$$\Delta\mathbf{w}_1 = (\mathbf{h}^\top\mathbf{w}_2 - \mathbf{h}^\top\mathbf{w}_1)\mathbf{w}_1$$

$$\Delta\mathbf{w}_2 = (\mathbf{h}^\top\mathbf{w}_1 - \mathbf{h}^\top\mathbf{w}_2)\mathbf{w}_2$$

We then observed the changes in logits for the original attribute $a$ ("Google") and the new attribute $a^*$ ("Apple") by incrementally adding each component to the residual stream $\mathbf{h}$, scaled by an interpolation factor $k \in [0, 1]$.

Fig. 7 provides a full breakdown of these effects for the edit (*"Chrome"*, *"was developed by"*, *"Apple"*). The results confirm our finding that the $\Delta\mathbf{w}_1$ component is effective at increasing the logit of the new attribute $a^*$, while the $\Delta\mathbf{w}_2$ component is effective at decreasing the logit of the old attribute $a$.

However, the plots also illustrate that $\mathbf{w}_1$ also suppresses the old attribute $a$, and $\mathbf{w}_2$ promotes the new attribute $a^*$, rather than each playing only a single role. This visual evidence reinforces the point that the components are not fully disentangled.

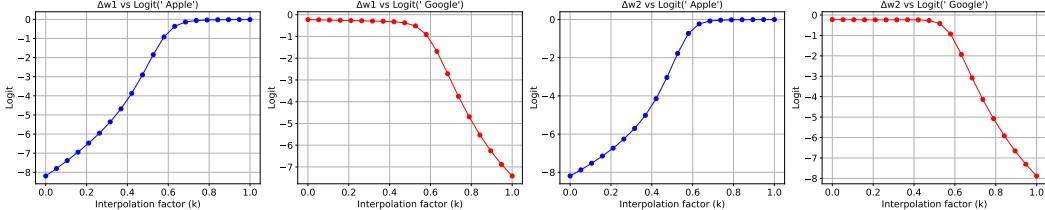

Figure 7: A full breakdown of the effects of applying scaled components $\Delta\mathbf{w}_1$ and $\Delta\mathbf{w}_2$ to the residual stream.

## G  CUMULATIVE ENERGY CURVES ACROSS MODELS

We analyze the spectral distribution of the key vector matrix $\mathbf{K}_{\text{entity}}$ defined in § 4.2. Fig. 8 visualizes the cumulative energy ratio of the singular values $\mathbf{S} = \text{diag}(\sigma_1, \ldots, \sigma_r)$ across the first four edited layers. The $x$-axis represents the rank index $k$, and the $y$-axis indicates the proportion of total energy explained by the top-$k$ components, calculated as $\sum_{i=1}^{k} \sigma_i^2 / E_{\text{total}}$. The red dots in the figure mark the "knee points," computed using the Kneedle algorithm (Satopaa et al., 2011), which represent the points of maximum curvature.

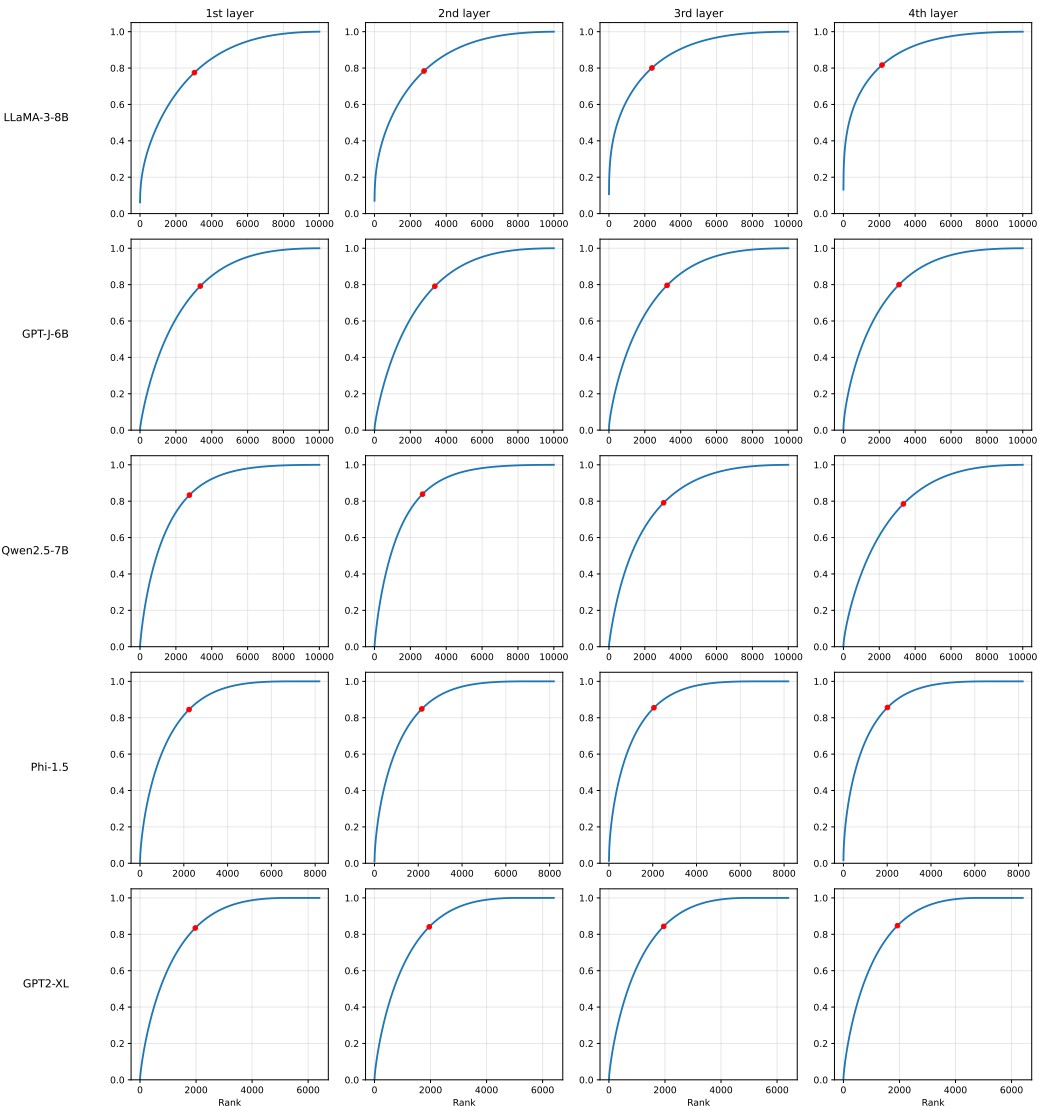

Figure 8: Cumulative energy curves of the layers targeted for editing across five models.

## H  PRE-COMPUTATION COST FOR BUILDING $\mathbf{K}_{\text{ENTITY}}$

This section reports the one-time pre-computation cost of constructing the entity matrix $\mathbf{K}_{\text{entity}}$ (defined in § 4.2) and running SVD, as a function of the number of entities $N$. All timings and memory measurements are obtained on an NVIDIA A100 80GB PCIe GPU. These steps are performed once per model–layer pair prior to editing; afterward, editing only requires loading the precomputed $(\mathbf{U}, \mathbf{S})$ factors and applying an energy-based truncation, which adds negligible overhead.

Table 15 shows that the dominant cost is computing the $k$ vectors used to form $\mathbf{K}_{\text{entity}}$, which scales approximately linearly with $N$. In contrast, the SVD time is comparatively small but still grows with $N$, and peak VRAM increases superlinearly in our setup (LLaMA3-Instruct (8B), averaged over five edited layers). To quantify sensitivity to $N$, we compare subspaces obtained with smaller values of $N$ to the reference subspace extracted from the full 25,799-entity set using the mean canonical correlation; Table 16 shows that similarity rises quickly with $N$ and already reaches 0.81 at $N$=10k, close to the full-data subspace.

Taken together, these results indicate that $N$=10,000 provides a favorable trade-off, substantially reducing one-time pre-computation time and peak memory compared to the full setting while producing a subspace that remains highly aligned with the full-data reference; consistent with this, editing performance using the $N$=10k subspace is nearly indistinguishable from performance using the full-entity subspace (Table 17).

Table 15: One-time pre-computation cost for building $\mathbf{K}_{\text{entity}}$ and running SVD as a function of $N$ (LLaMA3-Instruct (8B), five layers; NVIDIA A100 80GB PCIe).

| $N$ | Peak VRAM | $k$-vectors (s) | SVD (s) | Total (s) |
|---|---|---|---|---|
| 500 | 2.71 GB | 256.1 | 0.5 | 256.6 |
| 1K | 2.74 GB | 528.4 | 7.1 | 535.5 |
| 2K | 2.80 GB | 1059.3 | 2.8 | 1062.0 |
| 4K | 2.95 GB | 2245.5 | 10.9 | 2256.4 |
| 6K | 3.13 GB | 3405.9 | 29.5 | 3435.4 |
| 10K | 3.59 GB | 5906.3 | 120.6 | 6026.9 |
| 14K | 4.16 GB | 8043.0 | 333.3 | 8376.3 |
| 18K | 5.95 GB | 11139.7 | 402.6 | 11542.2 |
| 22K | 8.25 GB | 16520.9 | 501.7 | 17022.6 |
| 25.8K | 10.82 GB | 19565.8 | 589.2 | 20154.9 |

Table 16: Subspace similarity to the full-data reference as $N$ varies, measured by mean canonical correlation.

| Entities ($N$) | 500 | 1k | 2k | 4k | 6k | 10k | 14k | 18k | 22k | 25.8k (full) |
|---|---|---|---|---|---|---|---|---|---|---|
| Mean Canonical Corr. | 0.35 | 0.42 | 0.55 | 0.68 | 0.72 | 0.81 | 0.86 | 0.91 | 0.97 | 1.00 |

Table 17: Editing performance using the $N$=10k subspace vs. the full-entity subspace.

| Setting | $S$ | Eff. | Gen. | Spe. |
|---|---|---|---|---|
| $N = 10,000$ | 86.8 | 99.7 | 90.3 | 74.2 |
| Full ($N = 25,799$) | 87.4 | 99.4 | 88.6 | 77.2 |

# I  MEASURING ENTITY VS. CONTEXT EFFECTS IN $\mathcal{K}_s$ AND $\mathcal{K}_s^{\perp}$

To complement our variance-based subspace construction, we perform a 4-way cross-pair analysis to test how the entity-agnostic subspace $\mathcal{K}_s^{\perp}$ and the entity-specific subspace $\mathcal{K}_s$ respond to controlled changes in *entity identity* and *sentence context*. The goal is to disentangle entity effects from context effects using paired interventions.

We sample two entities $(E_1, E_2)$ and two contexts $(C_1, C_2)$, and construct four sentences:
$$\{S(E_1, C_1), \ S(E_1, C_2), \ S(E_2, C_1), \ S(E_2, C_2)\}.$$
For illustration, $C_1$ and $C_2$ can be instantiated so that the entity's syntactic role and position are fixed while only the *prefix context* changes, e.g., $C_1$: "*In the news article, $\{E\}$ was mentioned repeatedly.*" and $C_2$: "*During the committee meeting, $\{E\}$ was mentioned repeatedly.*"

Since we extract the key vector at the entity's last-token position in a decoder-only causal LM, the representation at that position depends only on the left context (prefix) before the entity. Therefore, context variation is applied to the prefix preceding $\{E\}$.

For each sentence, we extract the key vector $\mathbf{k}$ at the entity's last-token position and project it onto each subspace. Let $\mathcal{K} \in \{\mathcal{K}_s, \mathcal{K}_s^{\perp}\}$, and let $\mathcal{K}(E, C)$ denote the projected key vector obtained from $S(E, C)$ in subspace $\mathcal{K}$.

We then define two controlled changes. The *context-swap change* measures how much the representation changes when the entity is fixed and only the context is changed:
$$d_{\text{ctx}}^{\mathcal{K}} = \|\mathcal{K}(E_1, C_1) - \mathcal{K}(E_1, C_2)\|^2.$$

The *entity-swap change* measures how much the representation changes when the context is fixed and only the entity is changed. To reduce context-specific bias, we compute it under both contexts and average:

$$d_{\text{ent}}^{\mathcal{K}} = \tfrac{1}{2}\Big( \|\mathcal{K}(E_1, C_1) - \mathcal{K}(E_2, C_1)\|^2 + \|\mathcal{K}(E_1, C_2) - \mathcal{K}(E_2, C_2)\|^2 \Big).$$

Because $\mathcal{K}_s$ and $\mathcal{K}_s^{\perp}$ may have different norm scales, absolute distances alone can confound sensitivity with scale. We therefore also report a norm-normalized distance:

$$rd(a, b) = \frac{\|a - b\|^2}{\|a\|^2 + \|b\|^2}.$$

We define $rd_{\text{ctx}}^{\mathcal{K}}$ and $rd_{\text{ent}}^{\mathcal{K}}$ by replacing each squared-distance term in $d_{\text{ctx}}^{\mathcal{K}}$ and $d_{\text{ent}}^{\mathcal{K}}$ with $rd(\cdot, \cdot)$, respectively, so that the comparison reflects relative representational change rather than raw magnitude.

To summarize entity sensitivity, we compare entity-swap and context-swap changes using a context-baseline-subtracted gap:

$$\Delta^{\mathcal{K}} = d_{\text{ent}}^{\mathcal{K}} - d_{\text{ctx}}^{\mathcal{K}}, \qquad r\Delta^{\mathcal{K}} = rd_{\text{ent}}^{\mathcal{K}} - rd_{\text{ctx}}^{\mathcal{K}}.$$

These quantities measure how much additional change is induced by swapping the entity, beyond the change already caused by swapping the context; larger values indicate stronger entity sensitivity relative to context sensitivity.

All values are averaged over 1,000 randomly sampled 4-way cross-pairs. Both subspaces are more sensitive to entity identity than to contextual variation, but the effect is substantially stronger in $\mathcal{K}_s$. In $\mathcal{K}_s$, the entity-swap change clearly exceeds the context-swap change ($d_{\text{ent}}^{\mathcal{K}_s} \approx 18.09$ vs. $d_{\text{ctx}}^{\mathcal{K}_s} \approx 2.79$), yielding a large gap ($\Delta^{\mathcal{K}_s} \approx 15.30$). In contrast, $\mathcal{K}_s^{\perp}$ shows much smaller absolute variation ($d_{\text{ctx}}^{\mathcal{K}_s^{\perp}} \approx 0.42$, $d_{\text{ent}}^{\mathcal{K}_s^{\perp}} \approx 4.52$) and a smaller gap ($\Delta^{\mathcal{K}_s^{\perp}} \approx 4.09$). This pattern remains after norm normalization: the normalized gap is also much larger in $\mathcal{K}_s$ ($r\Delta^{\mathcal{K}_s} \approx 0.80$) than in $\mathcal{K}_s^{\perp}$ ($r\Delta^{\mathcal{K}_s^{\perp}} \approx 0.36$). Overall, these results support the interpretation that $\mathcal{K}_s$ amplifies entity-specific differences more strongly, whereas $\mathcal{K}_s^{\perp}$ is comparatively less entity-selective.

## J  ADDITIONAL HEATMAP VISUALIZATIONS

This appendix presents extended visualizations to complement the perturbation analysis in § 6.1. While the main text provides a representative example, we offer these additional examples to illustrate how the perturbation patterns manifest across different contexts.

We provide further examples from the Wikinews Article Dataset, illustrating the perturbation at each entity's last token position. For each of the five articles presented, we show the $L_2$ norm of the difference in the residual streams of the final edited layer between the original model and models edited using SUIT, MEMIT, and AlphaEdit.

The figures are presented in the order: SUIT, MEMIT, AlphaEdit. As visually represented by the color intensity, SUIT consistently reduces the perturbation on the last token of the entity compared to both MEMIT and AlphaEdit. The same color indicates the same amount of perturbation across all figures.

Prime minister Hari Kostov of Macedonia has resigned from his position as of Monday, November 15. The Macedonian parliament will meet on Thursday to decide whether or not to accept his resignation. The BBC quoted Kostov, who was appointed last May, to have said that "there is no will for genuine teamwork" within the coalition, and that one of the parties in the current government has been promoting corruption and nepotism. Kostov also claimed that the preoccupation with the rights of Albanian ethnic minority in Macedonia was obstructing economic modernization, according to Reuters. Kostov himself was not a member of any of the coalition's parties. Kostov's resignation was preceded by a referendum organized by the Macedonian opposition, aimed against a decentralisation law which would have given the Albanian ethnic minority in Macedonia additional rights. The referendum was declared null and void due to a low turnout. According to the NOS, some now fear that fights between Albanian guerrillas and the Macedonian army, which came to a halt in 2001, will start again.

Prime minister Hari Kostov of Macedonia has resigned from his position as of Monday, November 15. The Macedonian parliament will meet on Thursday to decide whether or not to accept his resignation. The BBC quoted Kostov, who was appointed last May, to have said that "there is no will for genuine teamwork" within the coalition, and that one of the parties in the current government has been promoting corruption and nepotism. Kostov also claimed that the preoccupation with the rights of Albanian ethnic minority in Macedonia was obstructing economic modernization, according to Reuters. Kostov himself was not a member of any of the coalition's parties. Kostov's resignation was preceded by a referendum organized by the Macedonian opposition, aimed against a decentralisation law which would have given the Albanian ethnic minority in Macedonia additional rights. The referendum was declared null and void due to a low turnout. According to the NOS, some now fear that fights between Albanian guerrillas and the Macedonian army, which came to a halt in 2001, will start again.

Prime minister Hari Kostov of Macedonia has resigned from his position as of Monday, November 15. The Macedonian parliament will meet on Thursday to decide whether or not to accept his resignation. The BBC quoted Kostov, who was appointed last May, to have said that "there is no will for genuine teamwork" within the coalition, and that one of the parties in the current government has been promoting corruption and nepotism. Kostov also claimed that the preoccupation with the rights of Albanian ethnic minority in Macedonia was obstructing economic modernization, according to Reuters. Kostov himself was not a member of any of the coalition's parties. Kostov's resignation was preceded by a referendum organized by the Macedonian opposition, aimed against a decentralisation law which would have given the Albanian ethnic minority in Macedonia additional rights. The referendum was declared null and void due to a low turnout. According to the NOS, some now fear that fights between Albanian guerrillas and the Macedonian army, which came to a halt in 2001, will start again.

Figure 9: Perturbation heatmaps for sample article 1.

The proposal from Dutch Minister of Justice Piet Hein Donner to strengthen the anti-blasphemy provisions of the criminal code has been rejected by a majority of the Tweede Kamer, the country's parliament. Donner put forth the proposal shortly after the murder of Dutch filmmaker Theo van Gogh, but denied that Van Gogh's death had anything to do with the proposal. NOS reported that Donner's own party, the Christian Democratic Appeal (CDA), supported his proposal. However, their two coalition partners — the People's Party for Freedom and Democracy (Volkspartij voor Vrijheid en Democratie, VVD) and Democrats 66 (D66) — announced they would not back the ban. The Labour Party, the largest opposition party, also refused to vote in favour. Without their support, the motion could not be passed. Consequently, NRC Handelsblad reports, Donner withdrew his proposal. Although the Dutch criminal code already makes blasphemy illegal, the law has only been enforced three times since the 1930s. The article in question states that anyone who ridicules a cleric or relic may be imprisoned for up to three months. According to Dutch broadcaster RTL, Member of Parliament Lousewies van der Laan (D66), will make a motion on November 17 to have the article removed entirely from the criminal code.

The proposal from Dutch Minister of Justice Piet Hein Donner to strengthen the anti-blasphemy provisions of the criminal code has been rejected by a majority of the Tweede Kamer, the country's parliament. Donner put forth the proposal shortly after the murder of Dutch filmmaker Theo van Gogh, but denied that Van Gogh's death had anything to do with the proposal. NOS reported that Donner's own party, the Christian Democratic Appeal (CDA), supported his proposal. However, their two coalition partners — the People's Party for Freedom and Democracy (Volkspartij voor Vrijheid en Democratie, VVD) and Democrats 66 (D66) — announced they would not back the ban. The Labour Party, the largest opposition party, also refused to vote in favour. Without their support, the motion could not be passed. Consequently, NRC Handelsblad reports, Donner withdrew his proposal. Although the Dutch criminal code already makes blasphemy illegal, the law has only been enforced three times since the 1930s. The article in question states that anyone who ridicules a cleric or relic may be imprisoned for up to three months. According to Dutch broadcaster RTL, Member of Parliament Lousewies van der Laan (D66), will make a motion on November 17 to have the article removed entirely from the criminal code.

The proposal from Dutch Minister of Justice Piet Hein Donner to strengthen the anti-blasphemy provisions of the criminal code has been rejected by a majority of the Tweede Kamer, the country's parliament. Donner put forth the proposal shortly after the murder of Dutch filmmaker Theo van Gogh, but denied that Van Gogh's death had anything to do with the proposal. NOS reported that Donner's own party, the Christian Democratic Appeal (CDA), supported his proposal. However, their two coalition partners — the People's Party for Freedom and Democracy (Volkspartij voor Vrijheid en Democratie, VVD) and Democrats 66 (D66) — announced they would not back the ban. The Labour Party, the largest opposition party, also refused to vote in favour. Without their support, the motion could not be passed. Consequently, NRC Handelsblad reports, Donner withdrew his proposal. Although the Dutch criminal code already makes blasphemy illegal, the law has only been enforced three times since the 1930s. The article in question states that anyone who ridicules a cleric or relic may be imprisoned for up to three months. According to Dutch broadcaster RTL, Member of Parliament Lousewies van der Laan (D66), will make a motion on November 17 to have the article removed entirely from the criminal code.

Figure 10: Perturbation heatmaps for sample article 2.

Secretary of State Colin Powell submitted his long-expected resignation as of Monday, November 15, and his resignation was accepted by President George W. Bush. His resignation letter was sent to the President on Friday. Powell has said that it was always his intention to serve only one term. The San Gabriel Valley Tribune said that Powell often had disputes with Bush Administration officials holding what the newspaper termed "more hawkish" views. On Tuesday, President Bush announced his nomination of National Security Advisor Dr. Condoleezza Rice as Powell's successor. Reuters cited senior Bush administration officials as saying that her deputy, Stephen Hadley, will succeed her in her role as Assistant to the President for National Security Affairs if she is confirmed as Secretary of State.

Secretary of State Colin Powell submitted his long-expected resignation as of Monday, November 15, and his resignation was accepted by President George W. Bush. His resignation letter was sent to the President on Friday. Powell has said that it was always his intention to serve only one term. The San Gabriel Valley Tribune said that Powell often had disputes with Bush Administration officials holding what the newspaper termed "more hawkish" views. On Tuesday, President Bush announced his nomination of National Security Advisor Dr. Condoleezza Rice as Powell's successor. Reuters cited senior Bush administration officials as saying that her deputy, Stephen Hadley, will succeed her in her role as Assistant to the President for National Security Affairs if she is confirmed as Secretary of State.

Secretary of State Colin Powell submitted his long-expected resignation as of Monday, November 15, and his resignation was accepted by President George W. Bush. His resignation letter was sent to the President on Friday. Powell has said that it was always his intention to serve only one term. The San Gabriel Valley Tribune said that Powell often had disputes with Bush Administration officials holding what the newspaper termed "more hawkish" views. On Tuesday, President Bush announced his nomination of National Security Advisor Dr. Condoleezza Rice as Powell's successor. Reuters cited senior Bush administration officials as saying that her deputy, Stephen Hadley, will succeed her in her role as Assistant to the President for National Security Affairs if she is confirmed as Secretary of State.

Figure 11: Perturbation heatmaps for sample article 3.

Investigators said that Saddam Hussein diverted money from the Oil-for-Food Program to pay millions of dollars to families of suicide bombers from the West Bank and Gaza Strip who carried out attacks on Israeli civilians. Paul Volcker, a former American official investigating the diverted funds scandal, has taken some heat from advocates demanding that he haul United Nations personnel before the US Congress. His reason for not subjecting them to this degree of open scrutiny is that it would have the perverse effect of pushing them into refusing to cooperate with the investigation at all. He plans to release documentary evidence early next year, when his investigation is complete.

Investigators said that Saddam Hussein diverted money from the Oil-for-Food Program to pay millions of dollars to families of suicide bombers from the West Bank and Gaza Strip who carried out attacks on Israeli civilians. Paul Volcker, a former American official investigating the diverted funds scandal, has taken some heat from advocates demanding that he haul United Nations personnel before the US Congress. His reason for not subjecting them to this degree of open scrutiny is that it would have the perverse effect of pushing them into refusing to cooperate with the investigation at all. He plans to release documentary evidence early next year, when his investigation is complete.

Investigators said that Saddam Hussein diverted money from the Oil-for-Food Program to pay millions of dollars to families of suicide bombers from the West Bank and Gaza Strip who carried out attacks on Israeli civilians. Paul Volcker, a former American official investigating the diverted funds scandal, has taken some heat from advocates demanding that he haul United Nations personnel before the US Congress. His reason for not subjecting them to this degree of open scrutiny is that it would have the perverse effect of pushing them into refusing to cooperate with the investigation at all. He plans to release documentary evidence early next year, when his investigation is complete.

Figure 12: Perturbation heatmaps for sample article 4.

A report by the United States government's Congressional Research Service (CRS) analysing al-Qaeda statements was made public Tuesday. The report examines methods used, ideas presented, and audience intended in al-Qaeda public statements, and how they have changed over time. The CRS is an auxiliary research office for the United States Congress, and does not make its unclassified reports public. The full report is available at the website of the Secrecy News project run by the Federation of American Scientists. Released November 16th, 2004, it is titled "Al Qaeda: Statements and Evolving Ideology".

A report by the United States government's Congressional Research Service (CRS) analysing al-Qaeda statements was made public Tuesday. The report examines methods used, ideas presented, and audience intended in al-Qaeda public statements, and how they have changed over time. The CRS is an auxiliary research office for the United States Congress, and does not make its unclassified reports public. The full report is available at the website of the Secrecy News project run by the Federation of American Scientists. Released November 16th, 2004, it is titled "Al Qaeda: Statements and Evolving Ideology".

A report by the United States government's Congressional Research Service (CRS) analysing al-Qaeda statements was made public Tuesday. The report examines methods used, ideas presented, and audience intended in al-Qaeda public statements, and how they have changed over time. The CRS is an auxiliary research office for the United States Congress, and does not make its unclassified reports public. The full report is available at the website of the Secrecy News project run by the Federation of American Scientists. Released November 16th, 2004, it is titled "Al Qaeda: Statements and Evolving Ideology".

Figure 13: Perturbation heatmaps for sample article 5.

