# OpenReview forum: "SUIT: Knowledge Editing with Subspace-Aware Key-Value Mappings"
_ICLR.cc/2026/Conference — ICLR 2026 Poster_

### Official Review · Reviewer_bGWe · 2025-10-28

**Soundness:** 3
**Presentation:** 2
**Contribution:** 2
**Rating:** 4
**Confidence:** 4

**Summary:**

This paper proposes SUIT (Subspace Knowledge Edit), a new knowledge editing method aimed at solving the problem of poor "specificity" (i.e., corrupting unrelated knowledge) that is common in locate-then-edit approaches. The method claims to achieve its goal via two core constraints: 1) When computing the "key vector" $k$, it identifies and removes an "entity-agnostic" subspace derived from 10,000 samples, purportedly isolating the edit to "entity-specific" features. 2) When computing the "residual vector" $\delta$, it forcibly constrains the optimization to a fixed two-dimensional subspace, defined by two vectors $w_1$ and $w_2$ intended to promote the new knowledge and suppress the old. Experimental results show the method improves specificity on the COUNTERFACT dataset for models like LLaMA-3.

**Strengths:**

1. This paper identifies the challenge of low specificity in knowledge editing.
2. The method demonstrates a significant specificity boost on the COUNTERFACT dataset based on a generation-based criterion.
3. The paper provides mechanistic evidence (e.g., token-level perturbations in residual streams) to validate the reduced perturbation.

**Weaknesses:**

1. The paper **assumes** the residual vector (the knowledge update) can be constrained to a two-dimensional subspace which lacks theoretical justification.
2. The subspace identification relies on an energy threshold hyperparameter $\tau_{energy}$ to which the model performance is **sensitive** as shown in Figure 5. The optimal $\tau_{energy}$ value obtained on COUNTERFACT may not be applicable to other datasets which adds an extra tuning burden for the method's application.
3. The paper selects $N=10,000$ subjects for the SVD pre-computation step which has no justification for this specific number nor any sensitivity analysis. It is unclear how robust the identified "entity-agnostic" subspace is; a different $N$ could potentially yield a different subspace. Furthermore, the computational cost of this SVD step scales directly with $N$, yet the authors fail to demonstrate that $N=10,000$ represents a reasonable trade-off between computational cost and the stability of the resulting subspace.
4. The analysis in $\S 6.2.1$ does not independently substantiate the semantic claim that $K_s^{\perp}$ is “entity-agnostic.” The subspace is defined via an SVD criterion, and the subsequent variance comparison merely reflects that construction rather than testing semantic invariance.
5. The second $\lambda_{KL}$ that appears in Appendix B should be weight decay $\lambda_{WD}$, which is likely a typo.

**Questions:**

See above.

---

> ### Author Response · Authors · 2025-11-23
>
> Thank you for taking the time and effort to review our paper. We appreciate your recognition of the Specificity challenge in knowledge editing, your noting of the strong Specificity gains of SUIT, and your highlighting of our mechanistic evidence validating the reduced perturbation. Below, we respond to the important questions and concerns you raised.
>
> ---
>
> **Q1.** The paper assumes the residual vector (the knowledge update) can be constrained to a two-dimensional subspace which lacks theoretical justification.
>
> **A1.** Thank you for raising this concern. We apologize if our assumption that the residual update can be constrained to a two-dimensional subspace seemed under-justified in the paper, and we appreciate the opportunity to clarify it further here.
> Prior studies on the Linear Representation Hypothesis and the representation of factual knowledge (Elhage et al., 2022; Hernandez et al., 2024; Park et al., 2024) have empirically demonstrated that model knowledge is often encoded within specific directions or subspaces. Building on this theoretical foundation, we hypothesized that model knowledge could be effectively altered by updating only along specific directions.
>
> Accordingly, we assumed that $w_1$ corresponds to the direction that increases the logits of the target object ($o^*$), while $w_2$ corresponds to the direction that decreases the logits of the original object ($o$). Based on this assumption, we calculated $\delta$ by constraining the update strictly to these directions.
> Furthermore, through the analysis in §6.2.2, we empirically demonstrated that $w_1$ and $w_2$ are indeed the dominant directions responsible for the knowledge edit.
>
> ---
> References
>
> Elhage, N., Hume, T., Olsson, C., Schiefer, N., Henighan, T., Kravec, S., Hatfield-Dodds, Z., Lasenby, R., Drain, D., Chen, C., Grosse, R., McCandlish, S., Kaplan, J., Amodei, D., Wattenberg, M., and Olah, C. Toy Models of Superposition. arXiv 2022.
>
> Hernandez, E., Sen Sharma, A., Haklay, T., Meng, K., Wattenberg, M., Andreas, J., Belinkov, Y., and Bau, D. Linearity of Relation Decoding in Transformer Language Models. ICLR 2024.
>
> Park, K., Choe, Y. J., and Veitch, V. The Linear Representation Hypothesis and the Geometry of Large Language Models. arXiv 2024.

---

> ### Author Response · Authors · 2025-11-23
>
> **Q2.** The subspace identification relies on an energy threshold hyperparameter $\tau_{\text{energy}}$ to which the model performance is sensitive as shown in Figure 5. The optimal $\tau_{\text{energy}}$ value obtained on COUNTERFACT may not be applicable to other datasets which adds an extra tuning burden for the method's application.
>
> **A2.**
> Thank you for raising this concern. To avoid any misunderstanding, our main results do not rely on dataset-specific tuning of $\tau_{\text{energy}}$. We fix a single value ($\tau_{\text{energy}}=0.40$), and use it across all datasets for each model; this already indicates that the method is not highly sensitive to the dataset or model choice.
>
> Regarding your concern about extra tuning burden, we note that similar hyperparameters are also used in other locate-then-edit baselines, so this is not unique to SUIT. For example, other baselines includes the regularization $\lambda_{\mathrm{KL}}$ and $\lambda_{\mathrm{WD}}$ (as described in Appendix B), and AlphaEdit uses nullspace_threshold (as described in Appendix D.2). Prior work likewise provides reasonable default values (e.g., $\mathrm{kl\\_factor}$ =  0.0625, $\mathrm{nullspace\\_threshold}$ =  0.02).
>
> Still, to further demonstrate that $\tau_{\text{energy}}$ is not sensitive around its optimal region, we ran an additional sensitivity study. We swept $\tau \in \{0.35, 0.40, 0.45\}$ across multiple models and datasets, and observed only marginal changes in Efficacy and Generalization (<1 point) alongside smooth, moderate increases in Specificity (≈1–4 points). This indicates that $\tau_{\text{energy}}$ behaves stably across models and datasets, supporting $\tau_{\text{energy}}=0.40$ as a robust default without imposing substantial extra tuning burden.
>
> **CounterFact**
>
> | Model   | τ    |  Eff |  Gen |  Spe |
> | :------ | :--- | ---: | ---: | ---: |
> | LLaMA3  | 0.35 | 99.7 | 90.6 | 71.1 |
> |         | 0.40 | 99.7 | 90.3 | 74.2 |
> |         | 0.45 | 99.7 | 89.5 | 74.8 |
> | GPT-J   | 0.35 | 98.2 | 93.1 | 62.3 |
> |         | 0.40 | 98.6 | 93.3 | 64.1 |
> |         | 0.45 | 98.7 | 92.7 | 65.7 |
> | Qwen2.5 | 0.35 | 99.4 | 86.8 | 73.5 |
> |         | 0.40 | 99.5 | 86.8 | 74.4 |
> |         | 0.45 | 99.6 | 87.5 | 74.9 |
>
> **zsRE**
>
> | Model   | τ    |  Eff |  Gen |  Spe |
> | :------ | :--- | ---: | ---: | ---: |
> | LLaMA3  | 0.35 | 95.6 | 85.8 | 68.3 |
> |         | 0.40 | 95.2 | 85.7 | 68.5 |
> |         | 0.45 | 94.4 | 85.6 | 70.2 |
> | GPT-J   | 0.35 | 99.6 | 89.2 | 98.9 |
> |         | 0.40 | 99.7 | 89.5 | 99.3 |
> |         | 0.45 | 99.7 | 89.4 | 99.8 |
> | Qwen2.5 | 0.35 | 93.9 | 76.3 | 96.2 |
> |         | 0.40 | 93.9 | 76.9 | 96.6 |
> |         | 0.45 | 93.8 | 76.9 | 97.7 |

---

> ### Author Response · Authors · 2025-11-23
>
> **Q3.** The paper selects N = 10,000 subjects for the SVD pre-computation step which has no justification for this specific number nor any sensitivity analysis. It is unclear how robust the identified "entity-agnostic" subspace is; a different N could potentially yield a different subspace. Furthermore, the computational cost of this SVD step scales directly with N, yet the authors fail to demonstrate that N = 10,000 represents a reasonable trade-off between computational cost and the stability of the resulting subspace.
>
> **A3.**
> We appreciate this important concern. Below, we provide sensitivity and cost analyses showing that N=10,000 is a practical and well-justified trade-off between subspace stability and pre-computation cost. The detailed results and analyses are provided below.
>
> *1. Computational cost*
>
> We would like to clarify that building the subject matrix $K_{\text{subj}}$ and running SVD are one-time pre-computations done once per model–layer pair, before editing that model. Hence, the additional overhead during editing is negligible, since it only involves loading $(U, S)$ and a simple energy-based truncation.
> As you noted, the one-time pre-computation cost scales roughly linearly with the number of subjects N, whereas peak VRAM increases superlinearly in our setup. We present the corresponding Llama-3-8B-Instruct results over five layers below.
>
> |     N | Peak VRAM | $k$ vector computation (s) | SVD computation (s) | Total (s) |
> | ----: | --------: | -------------------------: | ------------------: | --------: |
> |   500 |   2.71 GB |                      256.1 |                 0.5 |     256.6 |
> |    1K |   2.74 GB |                      528.4 |                 7.1 |     535.5 |
> |    2K |   2.80 GB |                     1059.3 |                 2.8 |    1062.0 |
> |    4K |   2.95 GB |                     2245.5 |                10.9 |    2256.4 |
> |    6K |   3.13 GB |                     3405.9 |                29.5 |    3435.4 |
> |   10K |   3.59 GB |                     5906.3 |               120.6 |    6026.9 |
> |   14K |   4.16 GB |                     8043.0 |               333.3 |    8376.3 |
> |   18K |   5.95 GB |                    11139.7 |               402.6 |   11542.2 |
> |   22K |   8.25 GB |                    16520.9 |               501.7 |   17022.6 |
> | 25.8K |  10.82 GB |                    19565.8 |               589.2 |   20154.9 |
>
> *2. Stability of the resulting subspace*
>
> To evaluate how sensitive the identified subspace is to the choice of N, we use the subspace extracted from all 25,799 subjects as a reference and compare it to subspaces obtained with smaller N. We measure similarity using mean canonical correlation. The similarity is low for small N (0.35 at N=500 and 0.42 at N=1,000), but increases quickly as more subjects are used, reaching 0.81 already at N=10,000.
>
> |         Subjects (N) |  500 |   1k |   2k |   4k |   6k |  10k |  14k |  18k |  22k | 25.8k (full) |
> | -------------------: | ---: | ---: | ---: | ---: | ---: | ---: | ---: | ---: | ---: | -----------: |
> | Mean Canonical Corr. | 0.35 | 0.42 | 0.55 | 0.68 | 0.72 | 0.81 | 0.86 | 0.91 | 0.97 |         1.00 |
>
> *3. Conclusion*
>
> Considering both computational cost and subspace stability, N=10,000 offers a favorable trade-off: compared to using the full set, it substantially reduces one-time pre-computation time by 70% and memory by 67% while still producing a subspace that is highly aligned with the full-data reference. To further validate this choice, we also find that editing performance using the N=10,000 subspace is nearly indistinguishable from performance using the full-subject subspace (see the result below).
>
> | N          |    S |  Eff |  Gen |  Spe |
> | ---------- | ---: | ---: | ---: | ---: |
> | N = 10,000 | 86.8 | 99.7 | 90.3 | 74.2 |
> | Full       | 87.4 | 99.4 | 88.6 | 77.2 |
>
> **This has been added to the revised submission and is provided in Appendix I “Pre-computation Cost for Building $\mathbf{K}_{\text{subject}}$”.**

---

> ### Author Response · Authors · 2025-11-23
>
> **Q4.** The analysis in §6.2.1 does not independently substantiate the semantic claim that $K_s^{\perp}$ is “entity-agnostic.” The subspace is defined via an SVD criterion, and the subsequent variance comparison merely reflects that construction rather than testing semantic invariance.
>
> **A4.**
> We would like to respectfully clarify that the analysis in §6.2.1 is not a circular verification based on the construction method, but rather a validation of generalization across different datasets.
> While it is true that we employed SVD to identify an "entity-agnostic" subspace, defined as a space where representations exhibit low variance across entities, we computed this subspace using the subjects from the PARAREL dataset (based on Wikidata). The significance of the variance analysis in §6.2.1 lies in the fact that we evaluated this pre-computed subspace on different datasets, specifically the CounterFact and zsRE benchmarks used for knowledge editing.
> In summary, the analysis was not merely re-confirming low variance on the data used for SVD (which would indeed be trivial). Instead, we aimed to verify whether the "entity-agnostic" property identified from PARAREL generalizes to the distinct entities found in knowledge editing tasks. The results empirically confirm that the subspace maintains low variance even for these unseen entities, thereby substantiating the claim that the subspace is truly "entity-agnostic".
>
> Additionally, to further support the semantic claim that $K_s^{\perp}$ is entity-agnostic beyond the variance-based SVD construction, we ran an auxiliary 4-way cross-pair experiment that disentangles entity effects from context effects. Concretely, we sample two entities $(E_1, E_2)$ and two contexts $(C_1, C_2)$, form four sentences ${S(E_1,C_1), S(E_1,C_2), S(E_2,C_1), S(E_2,C_2)}$, and extract the key vector $\mathbf{k}$ at each entity’s last token position in each sentence. The templates for $C_1$ and $C_2$ were drawn from the CounterFact dataset’s relation prompt templates (e.g., relation id P103 corresponds to "The mother tongue of {E} is").
> We project these vectors onto $K_s^{\perp}$ and $K_s$, and compare (i) within-entity changes (entity fixed, context swapped) and (ii) between-entity changes (context fixed, entity swapped). Then we compute pairwise distances between the projected vectors, and make them scale-invariant by using a norm-normalized distance. The entity effect is summarized as $r\Delta^{K} = rd_{between}^{K} - rd_{within}^{K}$, which measures how much more the subspace varies when the entity changes than when only the context changes. Under this normalization, the entity effect remains markedly stronger in $K_s$ than in $K_s^{\perp}$ ($r\Delta^{K_s}\approx 0.80$ vs. $r\Delta^{K_s^{\perp}}\approx 0.36$), i.e., over a 2x gap. All values are averaged over 1,000 randomly sampled cross-pairs. Overall, $K_s$ amplifies entity-specific differences far more strongly, consistent with its role as an entity-specific editing subspace.
> **We have added this analysis as Appendix J "Measuring Entity vs. Context Effects in $K_s$ and $K_s^{\perp}$" in the revised submission.**

---

> ### Author Response · Authors · 2025-11-23
>
> **Q5.** The second $\lambda_{\mathrm{KL}}$ that appears in Appendix B should be weight decay $\lambda_{\mathrm{WD}}$, which is likely a typo.
>
> **A5.** The second $\lambda_{\mathrm{KL}}$ in Appendix B is indeed a typo; it should be the weight decay term $\lambda_{\mathrm{WD}}$. We have corrected this throughout the revised submission. We apologize for any confusion this may have caused.

---

### Official Review · Reviewer_U4G2 · 2025-10-30

**Soundness:** 3
**Presentation:** 3
**Contribution:** 2
**Rating:** 4
**Confidence:** 3

**Summary:**

This paper introduces a novel knowledge editing method, SUIT (Subspace Knowledge Edit), which constrains updates to task-relevant subspaces by decomposing the key vector 𝑘 into entity-specific features and restricting the residual vector 𝛿 to the feature directions most relevant to the new object. By operating only within these targeted subspaces, SUIT enables precise and localized modification of factual knowledge.

**Strengths:**

1. The performance of SUIT is good and surpasses other baselines across datasets.

2. The idea of making a finer-grained distinction in the editing subspace is novel and also reminds me of SAE..

3. The paper's presentation is clear and well-organized.

**Weaknesses:**

1. It would be valuable to further evaluate the "ripple effects" metric of knowledge editing, which is mentioned in [1].

2. As with other locate-then-edit approaches, it would be good to explore whether this method can generalize beyond the triple-based question.

3. It would strengthen the work if experiments were extended to larger-scale models, such as 14B-parameter models.

4. On the ZSRE dataset, the average performance (S score) on two models is still lower than that of AlphaEdit.

5. SUIT generally underperforms AlphaEdit on the Generalization metric; it would be helpful to investigate the underlying reasons for this gap.

6. It would also be insightful to compare SUIT with SAE-based knowledge editing approaches and examine performance differences.

If the above weaknesses are addressed well, I will reconsider my rating.

---
**References**:

[1] Evaluating the Ripple Effects of Knowledge Editing in Language Models

**Questions:**

Please see the weaknesses list above.

---

> ### Author Response · Authors · 2025-11-23
>
> Thank you for taking the time and effort to review our paper. We appreciate your recognition of the novelty in our finer-grained subspace distinction and your note that the presentation is clear and well-organized. Below, we respond to the important questions and suggestions you raised.
>
> ---
> **Q1.** It would be valuable to further evaluate the "ripple effects" metric of knowledge editing, which is mentioned in [1].
>
> **A1.** Following the reviewer’s suggestion, we evaluated ripple effects using the RippleEdits benchmark. We follow the paper’s setup and test 100 single edits from the *Popular* split, which consists of high-frequency entities and is therefore higher-severity and more likely to trigger ripple errors. We report accuracy on the six RippleEdits criteria—Logical Generalization (LG), Compositionality I/II (CI/CII), Subject Aliasing (SA), Preservation (PV), and Relation Specificity (RS). The results are:
>
> | Method    |    LG |    CI |   CII |    SA |    PV |    RS |
> | --------- | ----: | ----: | ----: | ----: | ----: | ----: |
> | AlphaEdit | 0.210 | 0.482 | 0.000 | 1.000 | 1.000 | 0.597 |
> | SUIT      | 0.257 | 0.497 | 0.000 | 1.000 | 1.000 | 0.500 |
> | MEMIT     | 0.188 | 0.256 | 0.000 | 0.131 | 0.074 | 0.220 |
>
> Overall, SUIT clearly outperforms MEMIT on this challenging split and is broadly competitive with AlphaEdit. **We have added this result in Appendix F.7 "RippleEdits Evaluation Results" in the revised submission.**

---

> ### Author Response · Authors · 2025-11-23
>
> **Q2.** As with other locate-then-edit approaches, it would be good to explore whether this method can generalize beyond the triple-based question.
>
> **A2.** Thank you for the thought-provoking question. We believe that SUIT is not strictly constrained to triples, and its core idea can generalize to non-triple settings. In principle, it can be extended to unstructured edits by (i) expressing the target knowledge as a natural-language statement, (ii) locating the most influential token–layer site via causal tracing, and (iii) applying SUIT’s subspace-restricted update at that site. The current lack of standard benchmarks for unstructured editing prevents a quantitative study here. As you noted, generalizing beyond triple-based questions is a common concern for locate-then-edit approaches. This is definitely an exciting research direction and we leave a full evaluation of this extension to future work.

---

> ### Author Response · Authors · 2025-11-23
>
> **Q3.** It would strengthen the work if experiments were extended to larger-scale models, such as 14B-parameter models.
>
> **A3.** Following your suggestion, we ran additional experiments on a larger model, Llama2-13B. Since prior knowledge-editing baselines (e.g., MEMIT, AlphaEdit) report results and tuned hyperparameters only up to ~8B, there are no established baseline settings for models of this scale. For fairness, we therefore kept all methods’ hyperparameters identical to those for Appendix D.2 and evaluated them under the same sequential batch-editing setup (5 batches × 100 edits).
> Under this controlled setting, SUIT performs best overall, achieving the highest $S$ score with strong Specificity and improved Generalization compared to AlphaEdit. This indicates SUIT scales well to larger models.
>
> | Method    |   $S$ |  Eff |  Gen |  Spe |
> | --------- | ---: | ---: | ---: | ---: |
> | AlphaEdit | 79.3 | 98.4 | 76.5 | 68.4 |
> | SUIT      | 84.7 | 95.0 | 79.1 | 81.7 |
> | MEMIT     | 73.2 | 90.0 | 68.3 | 65.6 |
>
> **We have added these additional results to Appendix F.6 "Scaling to a Larger Model" in the revised submission.**

---

> ### Author Response · Authors · 2025-11-23
>
> **Q4.** On the ZSRE dataset, the average performance (S score) on two models is still lower than that of AlphaEdit.
>
> **A4.**
> This is mainly because SUIT does not stand out on zsRE’s under the standard Specificity metric. Below, we explain why this occurs and provide a more extensive, stricter analysis that better reflects Specificity preservation.
>
> *1. Limitations of Standard Specificity Measurement on zsRE*
>
> Prior work (e.g., MEMIT, AlphaEdit) evaluates zsRE’s Specificity with token-level partial credit against the dataset’s ground-truth answer. However, as shown below, the pre-edit model already scores very low, so post-edit changes are hard to attribute to an actual loss of Specificity.
>
> |           | LLaMA3 | GPT-J |
> | --------- | -------------------: | -------------------: |
> | Pre Edit  |                 32.0 |                 27.0 |
> | AlphaEdit |                 32.7 |                 28.2 |
> | SUIT      |                 32.7 |                 27.8 |
>
> To make zsRE’s Specificity metric more informative, we use the model’s own generated tokens, rather than the dataset’s ground truth, as the reference to preserve. Concretely, in the sequential setting, we record the model’s generated token sequence for each batch’s Specificity prompts immediately before applying that batch edit, and evaluate how much these answers change due to the batch. Table 1 is computed under this setup. **We have added these details to Appendix E.2.1 in the revised submission.**
>
>
> *2. Additional Specificity Results under a Stricter Test*
>
> The batch-level setting reported above and in the manuscript is still relatively lenient because it treats the model’s answer right before each batch as the ground truth, capturing only the incremental drift caused by that batch edit. Therefore, here we propose an additional, stricter evaluation setting. Specifically, we fix the original pre-edit answer as the ground truth and, after all edits, check whether its first 1 to 4 tokens are exactly preserved. We ran this evaluation on zsRE, and the results are shown in the table below. In this more rigorous evaluation setting, SUIT clearly preserves prior answers more reliably than AlphaEdit.
>
>
> | LLaMA3 | 1 token | 2 token | 3 token | 4 token |
> | --------- | ------: | ------: | ------: | ------: |
> | Pre Edit  |     100 |     100 |     100 |     100 |
> | AlphaEdit |    31.2 |    22.1 |    19.8 |    18.3 |
> | SUIT      |    51.9 |    43.2 |    41.2 |    39.8 |
>
>
>
> |    GPT-J    | 1 token | 2 token | 3 token | 4 token |
> | --------- | ------: | ------: | ------: | ------: |
> | Pre Edit  |     100 |     100 |     100 |     100 |
> | AlphaEdit |    94.1 |    93.8 |    68.8 |    65.3 |
> | SUIT      |    99.5 |    99.3 |    83.7 |    80.3 |
>
> **We have added these additional results to Appendix E.2.2 in the revised submission.**

---

> ### Author Response · Authors · 2025-11-23
>
> **Q5.** SUIT generally underperforms AlphaEdit on the Generalization metric; it would be helpful to investigate the underlying reasons for this gap.
>
> **A5.**
> SUIT’s lower Generalization than AlphaEdit in some settings can be viewed as a **trade-off between Generalization and Specificity**. Knowledge editing methods are typically evaluated by how well they balance Efficacy, Generalization, and Specificity, rather than by maximizing any single metric in isolation. Because of the trade-off among these metrics (particularly Specificity versus the other two) and our grid search for hyperparameters that optimize the $S$ metric, SUIT under this specific configuration may not outperform other baselines on all three metrics simultaneously.
>
> This balance is inherent to locate-then-edit methods and is inevitably shaped by hyperparameters, namely the regularization terms $\lambda_{\mathrm{KL}}$ and $\lambda_{\mathrm{WD}}$ explained in Appendix B, and AlphaEdit’s nullspace threshold described in Appendix D.2. In particular, AlphaEdit’s nullspace threshold adjusts the strength of the preservation constraint, with smaller thresholds tightening the constraint and typically increasing Specificity at a potential cost to Efficacy and Generalization.
>
> To illustrate how these hyperparameters shape performance in practice, we conducted controlled sweeps for both methods, with results reported in the tables below. These sweeps confirm that both SUIT and AlphaEdit can move along the Generalization–Specificity spectrum via their hyperparameters: AlphaEdit can raise Specificity by tightening nullspace threshold, whereas SUIT can favor Generalization by relaxing $\tau$. The key takeaway is that the **steepness of this trade-off** differs markedly across methods. In AlphaEdit, settings with higher Specificity result in substantial reductions in Efficacy and Generalization (e.g., LLaMA3 at nullspace threshold =0.004). In contrast, SUIT achieves large gains in Specificity as $\tau$ increases from 0.2 to 0.4 while Efficacy and Generalization remain almost unchanged for both LLaMA3 and GPT-J. Therefore, SUIT mitigates the Generalization–Specificity trade-off across hyperparameter values better than AlphaEdit. Across hyperparameter values, we can find a hyperparameter configuration where SUIT outperforms AlphaEdit on all the three metrics. For instance, as shown in the table below, for LLaMA3, SUIT (with $\tau$ = 0.4) outperforms AlphaEdit (with nullspace threshold = 0.02) across all metrics; for GPT-J, SUIT (with $\tau$ = 0.4) similarly outperforms AlphaEdit (with nullspace threshold = 0.08) across all metrics.
>
> **SUIT - LLaMA3**
>
> |        $\tau$ |  Eff |  Gen |  Spe |
> | ------------: | ---: | ---: | ---: |
> |           0.2 | 99.6 | 90.9 | 58.1 |
> |           0.3 | 99.6 | 90.6 | 65.9 |
> | 0.4 (Table 1) | 99.7 | 90.3 | 74.2 |
>
> **AlphaEdit - LLaMA3**
>
> | null space threshold |  Eff |  Gen |  Spe |
> | -------------------: | ---: | ---: | ---: |
> |                0.004 | 74.1 | 55.5 | 73.4 |
> |                0.001 | 95.9 | 82.1 | 54.6 |
> |       0.02 (Table 1) | 97.3 | 88.7 | 31.0 |
>
> **SUIT - GPT-J**
>
> |        $\tau$ |  Eff |  Gen |  Spe |
> | ------------: | ---: | ---: | ---: |
> |           0.2 | 98.6 | 94.0 | 61.9 |
> |           0.3 | 98.6 | 93.3 | 62.8 |
> | 0.4 (Table 1) | 98.6 | 93.3 | 64.1 |
>
> **AlphaEdit - GPT-J**
>
> | null space threshold |  Eff |  Gen |  Spe |
> | -------------------: | ---: | ---: | ----: |
> |                 0.08 | 98.4 | 91.5 | 53.97 |
> |                 0.01 | 98.4 | 93.8 | 52.41 |
> |       0.02 (Table 1) | 98.3 | 95.0 |  49.0 |

---

> ### Author Response · Authors · 2025-11-23
>
> **Q6.** It would also be insightful to compare SUIT with SAE-based knowledge editing approaches and examine performance differences.
>
> **A6.**
> After surveying SAE-based papers, we found that MicroEdit (Wang et al., 2025) is virtually the only approach that uses an SAE as a core mechanism for actual knowledge editing and reports results on standard factual KE benchmarks such as zsRE and CounterFact (other SAE-based works mainly use pretrained SAEs for inference-time steering or concept unlearning, not benchmarked factual edits; Zhao et al., 2025; Farrell et al., 2024).
> MicroEdit attaches a pretrained SAE to an intermediate residual stream, uses sparse activations to isolate a minimal set of necessary neurons for the target knowledge, and performs the edit by updating only the SAE decoder while keeping the LLM (and SAE encoder) frozen. Since MicroEdit is formulated as a one-fact-per-round editing procedure, extending it to large edit sets inevitably requires applying edits sequentially.
>
> Therefore, the tables below compare MicroEdit, AlphaEdit, and SUIT under the same sequential-edit setting, where facts are edited one at a time. We report performance after sequentially editing 10, 100, and 1,000 facts from the ConterFact dataset.
>
> **number of edits = 10**
>
> | Method    | $S$ |   Eff |   Gen |   Spe |
> | --------- | ----: | ----: | ----: | ----: |
> | MicroEdit |  55.1 |  90.0 |  30.0 | 100.0 |
> | AlphaEdit |  89.7 | 100.0 |  75.0 |  98.8 |
> | SUIT      | 100.0 | 100.0 | 100.0 | 100.0 |
>
> **number of edits = 100**
>
> | Method    |    $S$ |   Eff |  Gen |  Spe |
> | --------- | ---: | ----: | ---: | ---: |
> | MicroEdit | 32.0 |  47.0 | 16.0 | 99.0 |
> | AlphaEdit | 83.8 | 100.0 | 95.0 | 65.4 |
> | SUIT      | 96.9 | 100.0 | 95.5 | 95.4 |
>
> **number of edits = 1000**
>
> | Method    |    $S$ |  Eff |  Gen |  Spe |
> | --------- | ---: | ---: | ---: | ---: |
> | MicroEdit |  0.9 | 40.5 |  0.3 | 99.4 |
> | AlphaEdit | 36.5 | 96.5 | 90.2 | 16.5 |
> | SUIT      | 85.4 | 99.6 | 89.5 | 71.9 |
>
> Overall, compared to SUIT and AlphaEdit, MicroEdit shows poor Generalization even at 10 edits, and as the number of edits increases it increasingly fails to perform the edits effectively.
>
> (cf.) Table 1 in our paper reports 1,000 edits as 10 batches of 100, whereas the tables above use 1,000 strictly sequential single edits; even in this setting, SUIT still outperforms AlphaEdit.
>
> ---
> References
>
> Wang, S., Wang, Q., Niu, R., Kong, H., Chang, Y. MicroEdit: Neuron-level Knowledge Disentanglement and Localization in Lifelong Model Editing. EMNLP 2025.
>
> Zhao, Y., Devoto, A., Hong, G., Du, X., Gema, A. P., Wang, H., He, X., Wong, K.-F., Minervini, P. Steering Knowledge Selection Behaviours in LLMs via SAE-Based Representation Engineering (SpARE). NAACL 2025.
>
> Farrell, E., Lau, Y.-T., Conmy, A. Applying Sparse Autoencoders to Unlearn Knowledge in Language Models. arXiv 2024

---

### Official Review · Reviewer_anrX · 2025-11-01

**Soundness:** 3
**Presentation:** 3
**Contribution:** 2
**Rating:** 4
**Confidence:** 4

**Summary:**

This paper studies the unintended side effects of LLM knowledge editing and aims to improve specificity. Building on the locate-then-edit paradigm that treats the MLP down-projection as a linear associative memory W mapping keys to values, the authors propose Subspace Knowledge Edit (SUIT). SUIT (i) removes entity-agnostic components from the key vector by projecting onto and subtracting a data-driven “common” subspace obtained via SVD over many subject keys, and (ii) restricts the residual/value update to a low-dimensional subspace spanned by two optimized directions that respectively promote the new object and suppress the old one.

**Strengths:**

- Subspace isolation via data-driven SVD. The paper presents a clean and principled subspace isolation method: collect key vectors for a large set of subjects, perform SVD, and define the entity-agnostic subspace as the span of top singular vectors up to an energy threshold \tau; subtracting this projection yields an entity-specific key k. This is simple, effective, and empirically validated (variance analysis in Table 2; reduced interaction with the common subspace in Table 3)
- The experimental results suggest that the proposed method is comparable to SOTA baselines.

**Weaknesses:**

- Sensitivity to the energy threshold τenergy and selection cost. The core localization step relies on a manually chosen τenergy. Fig. 5 shows a nontrivial trade-off: as τenergy increases, generation tends to decrease while specificity increases, with a “sweet spot” around 0.3–0.4. Although efficacy is fairly stable, the turning point is not sharply defined and may require cross-validation that is data-dependent and potentially costly. Please (a) provide a principled selection rule/heuristic (e.g., knee detection on cumulative energy, variance ratio targets, or stability-based criteria), (b) report sensitivity across datasets/models when fixing \tau energy without extra tuning (you currently fix \tau=0.4 for all models ), and (c) discuss compute overhead for building K subject and running SVD (N=10k subjects) and how often this needs to be repeated per model/layer.
- Not uniformly state-of-the-art across all settings. While SUIT is very strong overall, it is not consistently the best on every model/dataset. For example, on ZSRE, SUIT’s harmonic mean S trails AlphaEdit slightly on GPT-J and Qwen (e.g., AlphaEdit S≈96.9 vs. SUIT S≈95.9 on GPT-J; AlphaEdit S≈89.6 vs. SUIT S≈88.2 on Qwen).
- AlphaEdit’s results are slightly lower than their original paper.

**Questions:**

Please see weaknesses.

---

> ### Author Response · Authors · 2025-11-23
>
> Thank you for taking the time and effort to review our paper. We appreciate your noting that our subspace isolation is simple, effective, and empirically supported by our analyses. Below, we respond to the thoughtful questions and concerns you raised.
>
> ---
>
> **Bullet 1(a).**
> Please provide a principled selection rule/heuristic (e.g., knee detection on cumulative energy, variance ratio targets, or stability-based criteria).
>
> **Answer to Bullet 1(a).**
>
> Our selection rule for this hyperparameter is as follows: we sweep the threshold and choose the value that maximizes the harmonic mean of the main performance metrics on a development split, which yields a value of 0.4. We then fix this single value across all models and datasets. **To make this clearer, we revised and refined the “Hyperparameter Tradeoff” subsection in §6.3 in the revised submission.** In addition, to justify that this choice is robust and principled, we (i) examined cumulative energy curves across multiple models and (ii) evaluated a knee-detection heuristic on these curves, following your suggestion, and found that it consistently selects a high-energy knee (τ ≈ 0.8) where performance degrades, showing that it is not suitable as a selection rule.
> The details of both analyses are presented below.
>
>
> To further justify fixing τ = 0.4 across all experiments, we plotted cumulative energy curves for all edited layers across five models: the three models from our main experiments and two additional models (Phi-1.5 and GPT-2 XL). The curve shapes are highly consistent across models, suggesting that energy accumulation follows a similar pattern across different model families. This consistency further supports using a shared heuristic range identified by the harmonic-mean criterion. **We have added these plots in the revised submission (Appendix H, “Cumulative Energy Curves Across Models,” Fig. 8).**
>
> We also investigated the 'knee detection' strategy on cumulative energy as you suggested. Our analysis across various models revealed that the knee point consistently falls near $ \tau \approx 0.8$. On the positive side, this reaffirms that different models show similar performance tendencies for varying hyperparameter values. On the negative side, however, as evidenced by Figure 5, performance significantly degrades at this range. Consequently, we determined that the knee detection method is not optimal for hyperparameter selection compared to the harmonic mean-based method.

---

> ### Author Response · Authors · 2025-11-23
>
> **Bullet 1(b).**
> Report sensitivity across datasets/models when fixing τ energy without extra tuning (you currently fix τ=0.4 for all models )
>
> **Answer to Bullet 1(b).**
>
> We report a sensitivity analysis of the energy threshold τ across datasets and models without any additional tuning. We vary τ ∈ {0.35, 0.40, 0.45} for LLaMA3, GPT-J, and Qwen2.5 on CounterFact and zsRE. Efficacy and Generalization are essentially stable across τ (≈< 1 point changes), while Specificity increases smoothly by ~1–4 pts as τ grows. Hence, τ in [0.35, 0.45] mainly controls Specificity.
>
> **CounterFact**
>
> | Model   | τ    |  Eff |  Gen |  Spe |
> | :------ | :--- | ---: | ---: | ---: |
> | LLaMA3  | 0.35 | 99.7 | 90.6 | 71.1 |
> |         | 0.40 | 99.7 | 90.3 | 74.2 |
> |         | 0.45 | 99.7 | 89.5 | 74.8 |
> | GPT-J   | 0.35 | 98.2 | 93.1 | 62.3 |
> |         | 0.40 | 98.6 | 93.3 | 64.1 |
> |         | 0.45 | 98.7 | 92.7 | 65.7 |
> | Qwen2.5 | 0.35 | 99.4 | 86.8 | 73.5 |
> |         | 0.40 | 99.5 | 86.8 | 74.4 |
> |         | 0.45 | 99.6 | 87.5 | 74.9 |
>
> **zsRE**
>
> | Model   | τ    |  Eff |  Gen |  Spe |
> | :------ | :--- | ---: | ---: | ---: |
> | LLaMA3  | 0.35 | 95.6 | 85.8 | 68.3 |
> |         | 0.40 | 95.2 | 85.7 | 68.5 |
> |         | 0.45 | 94.4 | 85.6 | 70.2 |
> | GPT-J   | 0.35 | 99.6 | 89.2 | 98.9 |
> |         | 0.40 | 99.7 | 89.5 | 99.3 |
> |         | 0.45 | 99.7 | 89.4 | 99.8 |
> | Qwen2.5 | 0.35 | 93.9 | 76.3 | 96.2 |
> |         | 0.40 | 93.9 | 76.9 | 96.6 |
> |         | 0.45 | 93.8 | 76.9 | 97.7 |

---

> ### Author Response · Authors · 2025-11-23
>
> **Bullet 1(c).**
> Discuss compute overhead for building K subject and running SVD (N=10k subjects) and how often this needs to be repeated per model/layer.
>
> **Answer to Bullet 1(c).**
> We would like to clarify that building the subject matrix $K_{\text{subj}}$ and running SVD are one-time pre-computations done once for each model layer, before editing that model. As in MEMIT and AlphaEdit, which precompute the $K_0$ matrix once and reuse it, we compute $K_{\text{subj}}$ once, cache the SVD factors ($U, S$), and reuse them for all edits on the same model. Hence, the additional overhead during editing is negligible, since it only involves loading ($U, S$) and a simple energy-based truncation.
> The upfront cost is moderate; for Llama-3-8B with N=10,000 subjects over 5 layers, the one-time pre-computation takes about 1.7 hours:
>
> | Peak VRAM | $k$ vector computation (s) | SVD computation (s) | Total (s) |
> | ----------------: | -------------------------: | ------------------: | --------: |
> |           3.59 GB |                     5906.3 |               120.6 |    6026.9 |
>
> **This has been added to the revised submission and is provided in Appendix I “Pre-computation Cost for Building $\mathbf{K}_{\text{subject}}$”.**

---

> ### Author Response · Authors · 2025-11-23
>
> **Bullet 2.**
> Not uniformly state-of-the-art across all settings. While SUIT is very strong overall, it is not consistently the best on every model/dataset. For example, on ZSRE, SUIT’s harmonic mean S trails AlphaEdit slightly on GPT-J and Qwen (e.g., AlphaEdit S≈96.9 vs. SUIT S≈95.9 on GPT-J; AlphaEdit S≈89.6 vs. SUIT S≈88.2 on Qwen).
>
>
> **Answer to Bullet 2.**
>
> We agree that SUIT is not uniformly the best on every model/dataset, and this is mainly because SUIT does not appear to stand out on zsRE under the standard Specificity metric. We explain below why this happens and provide more extensive, stricter results to better support the Specificity improvement.
>
> *1. Limitations of Standard Specificity Measurement on zsRE*
>
> Prior work (e.g., MEMIT, AlphaEdit) evaluates zsRE’s Specificity with token-level partial credit against the dataset’s ground-truth answer. However, as shown below, the pre-edit model already scores very low, so post-edit changes are hard to attribute to an actual loss of Specificity.
>
> |           | LLaMA3 | GPT-J |
> | --------- | -------------------: | -------------------: |
> | Pre Edit  |                 32.0 |                 27.0 |
> | AlphaEdit |                 32.7 |                 28.2 |
> | SUIT      |                 32.7 |                 27.8 |
>
> To make zsRE’s Specificity metric more informative, we use the model’s own generated tokens, rather than the dataset’s ground truth, as the reference to preserve. Concretely, in the sequential setting, we record the model’s generated token sequence for each batch’s Specificity prompts immediately before applying that batch edit, and evaluate how much these answers change due to the batch. Table 1 is computed under this setup. **We have added these details to Appendix E.2.1 in the revised submission.**
>
> *2. Additional Specificity Results under a Stricter Test*
>
> The batch-level setting reported above and in the manuscript is still relatively lenient because it treats the model’s answer right before each batch as the ground truth, capturing only the incremental drift caused by that batch edit. Therefore, here we propose an additional, stricter evaluation setting. Specifically, we fix the original pre-edit answer as the ground truth and, after all edits, check whether its first 1 to 4 tokens are exactly preserved. We ran this evaluation on zsRE, and the results are shown in the table below. In this more rigorous evaluation setting, SUIT clearly preserves prior answers more reliably than AlphaEdit.
>
>
> | LLaMA3 | 1 token | 2 token | 3 token | 4 token |
> | --------- | ------: | ------: | ------: | ------: |
> | Pre Edit  |     100 |     100 |     100 |     100 |
> | AlphaEdit |    31.2 |    22.1 |    19.8 |    18.3 |
> | SUIT      |    51.9 |    43.2 |    41.2 |    39.8 |
>
>
>
> |    GPT-J    | 1 token | 2 token | 3 token | 4 token |
> | --------- | ------: | ------: | ------: | ------: |
> | Pre Edit  |     100 |     100 |     100 |     100 |
> | AlphaEdit |    94.1 |    93.8 |    68.8 |    65.3 |
> | SUIT      |    99.5 |    99.3 |    83.7 |    80.3 |
>
> **We have added these additional results to Appendix E.2.2 in the revised submission.**

---

> ### Author Response · Authors · 2025-11-23
>
> **Bullet 3.**
> AlphaEdit’s results are slightly lower than their original paper.
>
> **Answer to Bullet 3.**
>
> We assume this concern refers to the CounterFact results. As noted in §5.1 (Evaluation Details), MEMIT and AlphaEdit appear lower because we evaluate the models on CounterFact using an evaluation setup that differs from prior reports and that we believe is more rigorous and practical.
> **First**, we restrict edits to facts the model answers correctly before editing (i.e., already-known facts), so the task measures genuine knowledge editing rather than inserting unknown facts. **Second**, we use a generation-based criterion: instead of relying only on probability checks such as $P(o^{\ast} \mid s, r) > P(o \mid s, r)$, we verify that the model’s actual generated answer changes from $o$ to $o^{*}$ after editing. For completeness, Appendix F.2 also reports results under the original probability-based criterion.
>
> We expect the reviewer’s concern about “low scores” to mainly relate to Specificity. We measure Specificity as preservation of facts the model truly knew before editing. Concretely, we keep only prompts where the pre-edit model actually generated the correct object (a 100% pre-edit-accurate subset) and then track how much accuracy drops on this subset after editing, giving a more faithful measure of knowledge preservation. Recent work has already warned that prior setups can overestimate preservation (Pohl et al., 2025; Yang et al., 2025; Zhang et al., 2024; 2025; Park et al., 2025). Our criteria therefore follow this critique to reduce overestimation, rather than arbitrarily departing from convention.
>
>
> ---
>
> References
>
> Pohl et al. Towards a Principled Evaluation of Knowledge Editors. ACL Workshop L2M2 2025.
>
> Yang et al. The Mirage of Model Editing: Revisiting Evaluation in the Wild. ACL 2025
>
> Zhang et al. A Comprehensive Study of Knowledge Editing for Large Language Models. arXiv 2024.
>
> Park et al. Context-Robust Knowledge Editing for Language Models. ACL 2025 Findings.

---

### Official Review · Reviewer_JqDj · 2025-11-03

**Soundness:** 2
**Presentation:** 3
**Contribution:** 2
**Rating:** 4
**Confidence:** 4

**Summary:**

This paper presents SUIT (Subspace Knowledge Edit), a new framework for editing factual knowledge in large language models while minimizing disruption to unrelated information. Building on the Linear Representation Hypothesis, the authors propose subspace-aware editing that confines changes to semantic subspaces relevant to the target entity and relation. SUIT improves upon prior methods like ROME, MEMIT, and AlphaEdit by refining the computation of the key and residual vectors through subspace decomposition. This design enhances specificity and stability during edits. Experiments on LLaMA-3-8B, GPT-J-6B, and Qwen2.5-7B show large gains in specificity (e.g., +43.2 points on LLaMA-3-8B) while maintaining strong edit accuracy.

**Strengths:**

This paper makes a solid and coherent contribution to the field of knowledge editing in large language models. It stands out in three main aspects.
(1) It provides a clear theoretical framing grounded in the Linear Representation Hypothesis, establishing a principled link between semantic subspaces and factual updates. This theoretical foundation not only motivates the method design but also enhances interpretability and conceptual rigor.
(2) The paper offers comprehensive and convincing analyses, including ablation studies, perturbation comparisons, and examinations of key components such as subspace isolation and feature decomposition. These analyses provide both empirical depth and transparency, showing that the improvements are consistent and well-supported.
(3) The overall structure and presentation are highly polished. The logical flow—from motivation to methodology, experiments, and analysis—is clear and well-balanced, and the exposition makes complex ideas accessible without oversimplifying them. Collectively, these qualities make the work both theoretically meaningful and practically persuasive.

**Weaknesses:**

While the paper presents a clear and well-executed framework, several aspects limit its novelty and clarity.
(1) The core technical innovation is not particularly strong. Although the subspace-based approach is conceptually sound, its methodological route appears closely aligned with the null-space perturbation ideas introduced in AlphaEdit. The distinction between the proposed subspace isolation and existing null-space constraints is not sufficiently emphasized, making the contribution seem more incremental than foundational.
(2) There are issues in the main experiments that reduce confidence in the reported improvements. The primary evaluation table introduces the “S” metric as the harmonic mean of three scores, yet the performance gain seems largely driven by unusually high specificity, while the other two components remain modest. Moreover, the specificity scores for baseline methods such as MEMIT and AlphaEdit appear unexpectedly low compared to prior results. Additionally, the table includes a “GA” metric that lacks explanation—it is likely intended to be “GC,” but this ambiguity should be clarified.
(3) Finally, although the appendix provides numerous additional analyses, their purpose and logical role within the overall argument are unclear. The paper does not clearly justify why these analyses are necessary or how they support the main claims. Some component analyses could also be presented more transparently to better highlight their individual contributions to the overall performance gains.

**Questions:**

(1) The proposed subspace-based approach seems conceptually similar to the null-space technique used in AlphaEdit. Could the authors elaborate on the fundamental distinction between the proposed subspace isolation and the existing null-space constraint formulation? In particular, what new insights or advantages does SUIT provide beyond current research’s framework?
(2) Regarding the main experimental results, the “S” metric—described as the harmonic mean of three sub-metrics—appears to be driven primarily by the high specificity score. Could the authors provide a detailed breakdown of the three components and explain whether the improvements are consistent across all of them? Additionally, the baseline scores for MEMIT and AlphaEdit seem unexpectedly low compared to prior reports. Were there any differences in implementation or evaluation setup that could explain this? Finally, please clarify the meaning of the “GA” metric in the main table; should this instead refer to “GC”?
(3) The additional analyses in the appendix are extensive, but their motivation is not entirely clear. Could the authors better articulate how each analysis (e.g., perturbation visualizations, subspace decomposition tests, or ablations) contributes to the main argument? For instance, which specific results directly support the claimed benefits of subspace isolation? More explicit linking between these analyses and the core findings would strengthen the logical flow of the paper.

---

> ### Author Response · Authors · 2025-11-23
>
> Thank you for taking the time and effort to review our paper. We are glad that you found our contribution solid and coherent, especially the theoretical grounding, comprehensive analyses, and overall clarity of presentation. Below, we respond to the main questions and concerns you raised.
>
> ---
>
> **Q1.** The proposed subspace-based approach appears conceptually close to AlphaEdit’s null-space technique. Could you clarify the fundamental differences between your subspace isolation and the standard null-space constraint formulation? Specifically, what new insights or practical advantages does SUIT offer beyond the existing framework?
>
> **A1.**
> At a high level, the fundamental difference between SUIT and AlphaEdit lies in **where** the subspace constraint is applied and **how** it is enforced. AlphaEdit constrains the row space of the update matrix $\\Delta$ to lie in the null space of $K_0$, directly restricting the update itself. Although this is surely a substantial improvement over prior approaches, it still leaves a high degree of freedom in parameter editing and leaves large room for model perturbation, as evidenced by relatively lower Specificity scores compared to our method.
>
> SUIT has a similar objective to AlphaEdit and many other on-going studies in terms of restricting the edited space. But SUIT proposes a more advanced method for identifying narrower subspaces for editing that can substantially prevent model perturbation by introducing a new perspective, i.e., building on well-established theoretical frameworks (e.g., the Linear Representation Hypothesis), which have received relatively little attention in the knowledge editing literature. Specifically, SUIT constrains the $\\mathbf{k}$ vector and $\\boldsymbol{\\delta}$ vector to an edit-relevant subspace. Grounded in the Linear Representation Hypothesis, we hypothesize that directions corresponding to the fact being edited can be localized to a subspace, and we restrict the computation of $\\mathbf{k}$ and $\\boldsymbol{\\delta}$ to that subspace so that they capture only edit-relevant components. Thus, SUIT operates at a deeper level than AlphaEdit: it refines the representations ($\\mathbf{k}, \\boldsymbol{\\delta}$) from which $\\Delta$ is computed. Building on MEMIT and AlphaEdit, SUIT explicitly formulates knowledge edits as operations confined to an edit-relevant hidden-state subspace, rather than acting over the full representation space.
>
> The main distinctions can be summarized as follows.
>
> | Method    | Objective                                                                                  | Scope                                                                                                 |
> | --------- | ------------------------------------------------------------------------------------------ | ----------------------------------------------------------------------------------------------------- |
> | AlphaEdit | Implement the preserved-knowledge term in MEMIT more strongly via a null-space constraint. | The update weight matrix $\\Delta$ ($W' = W + \\Delta$, where $W'$ is the weight matrix after editing). |
> | SUIT      | Identify and use only the subspace that is important for the edit.                         | The hidden-state vectors $\\mathbf{k}$ and $\\boldsymbol{\\delta}$ that encode the edited fact.          |
>
> So, in summary and to answer your question directly, our **key insight** is that the factual knowledge to be edited can reside in a specific edit-relevant subspace within the model’s hidden states. We demonstrate this by explicitly isolating that subspace in SUIT and applying the update only within it. This yields a **practical advantage** because, by confining the construction of $\\mathbf{k}$ and $\\boldsymbol{\\delta}$ to the isolated subspace, SUIT performs safer edits with reduced unintended perturbation, thereby preserving Specificity and General Capabilities much better in practice.

---

> ### Author Response · Authors · 2025-11-23
>
> **Q2.1.** Regarding the main experimental results, the “$S$” metric—described as the harmonic mean of three sub-metrics—appears to be driven primarily by the high specificity score. Could the authors provide a detailed breakdown of the three components and explain whether the improvements are consistent across all of them?
>
> **A2.1.**
> The $S$ metric follows prior knowledge-editing work such as ROME and MEMIT. It is defined as the harmonic mean of Efficacy, Generalization, and Specificity, which Table 1 also reports individually. Because knowledge editing often involves a **trade-off between Generalization and Specificity**, $S$ is a standard single-number summary of **how well a method balances Efficacy, Generalization, and Specificity**.
>
> Because of the trade-off among these metrics (particularly Specificity vs. the other two) and the grid search for hyperparameters that optimize for the $S$ metric, the resulting performance of SUIT under this specific hyperparameter configuration may not outperform other baselines on all three metrics simultaneously. That said, Table 1 shows that Efficacy and Generalization remain comparable to strong baselines (MEMIT, AlphaEdit), while Specificity improves consistently across most models and datasets, yielding a higher $S$ without sacrificing the other two axes. At the same time, Figure 5 shows that the hyperparameters $\tau_{\text{energy}}$ and $\lambda$ offer an explicit and interpretable control over this balance, with smaller values generally improving Generalization and larger values strengthening Specificity. This makes it possible in practice to tune SUIT either toward more uniformly strong Efficacy, Generalization, and Specificity, or to deliberately prioritize the component most aligned with a user’s application needs.
>
> To verify this, we conducted controlled sweeps for both methods (the nullspace threshold hyperparameter for AlphaEdit and $\tau_{\text{energy}}$ for SUIT). The results are summarized in the tables below. The table shows that certain hyperparameter configurations (e.g., the nullspace threshold of 0.08 and $\tau_{\text{energy}}=0.4$ for GPT-J) lead SUIT to outperform AlphaEdit for all the metrics simultaneously.
> In addition, these sweeps confirm that both SUIT and AlphaEdit can move along the Generalization–Specificity spectrum via their hyperparameters: AlphaEdit can raise Specificity by tightening its nullspace threshold, whereas SUIT can favor Generalization by relaxing $\tau_{\text{energy}}$ and increase Specificity by raising it. The key takeaway is that **the steepness of this trade-off** differs markedly across methods. Because SUIT can increase Specificity with much smaller impact on Efficacy and Generalization, its higher $S$ scores reflect a more favorable balance across all three metrics rather than being driven by Specificity alone.
>
> **SUIT - LLaMA3**
>
> |        $\tau$ |  Eff |  Gen |  Spe |
> | ------------: | ---: | ---: | ---: |
> |           0.2 | 99.6 | 90.9 | 58.1 |
> |           0.3 | 99.6 | 90.6 | 65.9 |
> |           0.4 | 99.7 | 90.3 | 74.2 |
>
> **AlphaEdit - LLaMA3**
>
> | null space threshold |  Eff |  Gen |  Spe |
> | -------------------: | ---: | ---: | ---: |
> |                0.004 | 74.1 | 55.5 | 73.4 |
> |                0.001 | 95.9 | 82.1 | 54.6 |
> |                 0.02 | 97.3 | 88.7 | 31.0 |
>
> **SUIT - GPT-J**
>
> |        $\tau$ |  Eff |  Gen |  Spe |
> | ------------: | ---: | ---: | ---: |
> |           0.2 | 98.6 | 94.0 | 61.9 |
> |           0.3 | 98.6 | 93.3 | 62.8 |
> |           0.4 | 98.6 | 93.3 | 64.1 |
>
> **AlphaEdit - GPT-J**
>
> | null space threshold |  Eff |  Gen |  Spe |
> | -------------------: | ---: | ---: | ----: |
> |                 0.08 | 98.4 | 91.5 | 54.0 |
> |                 0.01 | 98.4 | 93.8 | 52.4 |
> |                 0.02 | 98.3 | 95.0 |  49.0 |

---

> ### Author Response · Authors · 2025-11-23
>
> **Q2.2.** Additionally, the baseline scores for MEMIT and AlphaEdit seem unexpectedly low compared to prior reports. Were there any differences in implementation or evaluation setup that could explain this?
>
> **A2.2.**
>
> ### [CounterFact Dataset Evaluation]
>
> We first explain the CounterFact results. As noted in §5.1 (Evaluation Details), MEMIT and AlphaEdit appear lower because we evaluate the models on CounterFact using an evaluation setup that differs from prior reports and that we believe is more rigorous and practical.
> **First**, we restrict edits to facts the model answers correctly before editing (i.e., already-known facts), so the task measures genuine knowledge editing rather than inserting unknown facts. **Second**, we use a generation-based criterion: instead of relying only on probability checks such as $P(o^{\ast} \mid s, r) > P(o \mid s, r)$, we verify that the model’s actual generated answer changes from $o$ to $o^{*}$ after editing. For completeness, Appendix F.2 also reports results under the original probability-based criterion.
>
> We expect the reviewer’s concern about “low scores” to mainly relate to Specificity. We measure Specificity as preservation of facts the model truly knew before editing. Concretely, we keep only prompts where the pre-edit model actually generated the correct object (a 100% pre-edit-accurate subset) and then track how much accuracy drops on this subset after editing, giving a more faithful measure of knowledge preservation. Recent work has already warned that prior setups can overestimate preservation (Pohl et al., 2025; Yang et al., 2025; Zhang et al., 2024; 2025; Park et al., 2025). Our criteria therefore follow this critique to reduce overestimation, rather than arbitrarily departing from convention.
>
> ---
>
> References
>
> Pohl et al. Towards a Principled Evaluation of Knowledge Editors. ACL Workshop L2M2 2025.
>
> Yang et al. The Mirage of Model Editing: Revisiting Evaluation in the Wild. ACL 2025
>
> Zhang et al. A Comprehensive Study of Knowledge Editing for Large Language Models. arXiv 2024.
>
> Park et al. Context-Robust Knowledge Editing for Language Models. ACL 2025 Findings

---

> > ### Author Response · Authors · 2025-11-23
> >
> > **A2.2. Additional Explanation and Experimental Results**
> >
> > ### [zsRE Dataset Evaluation]
> >
> > While the above explanation concerned CounterFact, we now describe our zsRE evaluation. Following prior work, we adopt a generation-based setup that gives partial credit to gold tokens (details in Appendix E.2). We also tried applying the same pre-knowledge filtering as in CounterFact by restricting evaluation to facts the model answered correctly before editing, but this proved infeasible on zsRE, so we omit it and explain the reason in detail below.
> >
> > *1. Reason for not restricting zsRE to known facts*
> >
> > This is mainly because of zsRE’s question format. Answers often start with function words rather than the object, so the first token is frequently unrelated to the target (e.g., “the” 4,902, “a” 445, “he” 388, “she” 89, etc.). The model also tends to generate full-sentence responses instead of short object spans (e.g., $(s,r)$: “What airport is China United Airlines part of?”, gold $(o)$: “Beijing Nanyuan Airport,” model output: “China United Airlines is part of Beijing Capital International Airport.”). **We have added these details to Appendix D.3 in the revised submission.**
> >
> > *2. Limitations of Standard Specificity Measurement on zsRE*
> >
> > For zsRE, prior work (e.g., MEMIT, AlphaEdit) evaluates Specificity with token-level partial credit against the ground-truth answer. As shown below, the pre-edit model already scores very low, so post-edit changes are hard to attribute to an actual loss of Specificity.
> >
> > |           | LLaMA3 | GPT-J |
> > | --------- | -------------------: | -------------------: |
> > | Pre Edit  |                 32.0 |                 27.0 |
> > | AlphaEdit |                 32.7 |                 28.2 |
> > | SUIT      |                 32.7 |                 27.8 |
> >
> > To make zsRE’s Specificity metric more informative, we use the model’s own generated tokens, rather than the dataset’s ground truth, as the reference to preserve. Concretely, in the sequential setting, we record the model’s generated token sequence for each batch’s Specificity prompts immediately before applying that batch edit, and evaluate how much these answers change due to the batch. Table 1 is computed under this setup. **We have added these details to Appendix E.2.1 in the revised submission.**
> >
> > *3. Additional Specificity Results under a Stricter Test*
> >
> > The batch-level setting reported above and in the manuscript is still relatively lenient because it treats the model’s answer right before each batch as the ground truth, capturing only the incremental drift caused by that batch edit. Therefore, here we propose an additional, stricter evaluation setting. Specifically, we fix the original pre-edit answer as the ground truth and, after all edits, check whether its first 1 to 4 tokens are exactly preserved. We ran this evaluation on zsRE, and the results are shown in the table below. In this more rigorous evaluation setting, SUIT clearly preserves prior answers more reliably than AlphaEdit.
> >
> > | LLaMA3 | 1 token | 2 token | 3 token | 4 token |
> > | --------- | ------: | ------: | ------: | ------: |
> > | Pre Edit  |     100 |     100 |     100 |     100 |
> > | AlphaEdit |    31.2 |    22.1 |    19.8 |    18.3 |
> > | SUIT      |    51.9 |    43.2 |    41.2 |    39.8 |
> >
> >
> >
> > |    GPT-J    | 1 token | 2 token | 3 token | 4 token |
> > | --------- | ------: | ------: | ------: | ------: |
> > | Pre Edit  |     100 |     100 |     100 |     100 |
> > | AlphaEdit |    94.1 |    93.8 |    68.8 |    65.3 |
> > | SUIT      |    99.5 |    99.3 |    83.7 |    80.3 |
> >
> > **We have added these additional results to Appendix E.2.2 in the revised submission.**

---

> ### Author Response · Authors · 2025-11-23
>
> **Q2.3.** Finally, please clarify the meaning of the “GA” metric in the main table; should this instead refer to “GC”?
>
> **A2.3.** In Table 1, “GA” is a typo and should be “GC (General Capability)”. We have corrected this throughout the revised submission. We apologize for any confusion this may have caused.

---

> ### Author Response · Authors · 2025-11-23
>
> **Q3.** The additional analyses in the appendix are extensive, but their motivation is not entirely clear.
> Could the authors better articulate how each analysis (e.g., perturbation visualizations, subspace decomposition tests, or ablations) contributes to the main argument?
> For instance, which specific results directly support the claimed benefits of subspace isolation? More explicit linking between these analyses and the core findings would strengthen the logical flow of the paper.
>
> **A3.**
> Thank you for your careful reading. We apologize that the motivation and connection of the appendix analyses to our main argument were not stated clearly enough in the paper. The analyses you referred to correspond to Appendix G, Appendix H, and the ablation study in §6.3. Below, we explain how each analysis ties to the main text and directly supports our core claims. **In the revision, we have clarified this by beginning each of these sections with the corresponding part of the main text.**
>
> ### Appendix G
>
> This section provides a detailed analysis of our learned directions, empirically validating our core assumption: the calculation of $\delta$ should be constrained to direction relevant to the knowledge edit ($w_1$ increasing the logits of the target object $o^{\ast}$, and $w_2$ decreasing the logits of the original object $o$).
> In §6.2.2, we numerically demonstrated that our learned components ($w_1, w_2$) align more closely with the target knowledge than existing methods. While we showed that $w_1$ primarily boosts $o^{\ast}$ and $w_2$ suppresses $o$, our analysis also revealed that their roles are not perfectly disentangled; rather, both directions simultaneously increase $o^{\ast}$ and decrease $o$. Appendix G visually shows this finding by plotting the logit changes, offering intuitive evidence of this behavior within the learned subspace.
>
> ### Appendix H
>
> In §6.2.1, we argued that SUIT induces significantly lower perturbation compared to other methods, particularly at the last token of the entity where factual information is aggregated. Appendix H serves to visually substantiate this claim. By providing perturbation heatmaps for various random texts, we offer an intuitive visualization of where and how much perturbation occurs. These additional heat maps confirm that our method maintains high locality by minimizing unintended side effects on the unedited knowledge.
>
> ### §6.3 Ablation Study
>
> While our methodology stems from a unified motivation of subspace-constrained edits for both key vectors and value vectors, it consists of two distinct strategies applied to each $k$ and $\delta$. The ablation study in §6.3 was conducted to demonstrate that the superior performance of SUIT is not driven by a single factor but is the result of the synergy between these two components. As shown in the main text, the results confirm that combining both strategies is essential to achieving optimal performance.

---

### Author Response · Authors · 2025-11-28

Dear Reviewers,

Thank you very much again for taking the time to review our paper.
We have conducted extensive additional experiments to address all of your comments and posted the results and clarifications in the author comments 4 days ago.
We have also incorporated these updates into a revised manuscript, which has now been uploaded.

We would be very grateful if you could take a look and let us know at your convenience whether our answers adequately address your concerns.
Please let us know if you have any further questions or concerns.

Thank you.

---

### Author Response · Authors · 2025-12-03
**Review Summary**

Dear Area Chair,

We have carefully addressed all questions and concerns raised by the reviewers, extensively and faithfully conducting **all the additional experiments and analyses requested by the reviewers**. We have incorporated the results into the revised manuscript. While we posted our rebuttal early in the discussion period, we did not receive follow-up replies and, under the current process, had no further opportunity for interaction. Since it may be difficult to see from the rebuttal and reviews alone how thoroughly these concerns were addressed, we therefore provide below a concise overview of how our responses and new experiments directly address the reviewers’ main points.

- **Energy-threshold hyperparameter**: Reviewers raised two questions about the energy threshold $\tau$: how to choose $\tau$ in a principled way (anrX) and whether a single value can be reused across datasets and models (anrX, bGWe). For the first point, we clarify that on LLaMA-3-8B we sweep $\tau$ and set $\tau = 0.4$ to maximize the harmonic-mean score $\mathcal{S}$. For the second, while Table 1 already shows that this value transfers well across models and datasets, we further conducted additional experiments to sweep $\tau$ around 0.4 for each model–dataset pair to verify its robustness. We observed only mild variation, further validating the stability of this value. In addition, we conducted an empirical analysis of cumulative energy curves across five different models and found that they exhibit highly similar behavior, providing further validation that $\tau \approx 0.4$ is a broadly applicable setting.

- **$K_{\text{subj}}$ + SVD computation cost and the choice of $N$**: Reviewers raised concerns about (i) the computational cost of the $K_{\text{subj}}$ + SVD computation (anrX, bGWe) and (ii) the justification for using $N = 10{,}000$ subjects for $K_{\text{subj}}$ (bGWe). We clarified that building $K_{\text{subj}}$ and running SVD is a one-time pre-computation (e.g., about 1.7 hours for LLaMA-3-8B); once this is done, subsequent edits simply load the cached $(U, S)$. To justify $N = 10{,}000$, we performed an additional analysis in which we examined choices of $N$ ranging from 500 up to the full subject set (25.8K). For each setting, we compared the resulting subspace with that from the full set using canonical correlation analysis, jointly assessing pre-computation cost and editing performance. This analysis shows that $N = 10{,}000$ achieves essentially the same functional behavior as the full set while incurring substantially lower computational cost.

- **Comparison with SAE-Based Method**: Reviewer U4G2 requested we compare SUIT with an SAE-based editor. Hence, we conducted additional experiments with MicroEdit and compared with SUIT. We found that, as the number of edits increases, SUIT maintains substantially higher performance while MicroEdit degrades rapidly.

- **Riffle Effects**: As requested by Reviewer U4G2, we conducted additional experiments on ripple effects and evaluated SUIT on the RippleEdits benchmark. We found that it achieves clearly better ripple stability than MEMIT while remaining competitive with AlphaEdit.

- **Model Scale**: Reviewer U4G2 was concerned that SUIT had been evaluated on relatively small models. To address this concern, we conducted additional experiments to evaluate SUIT with a larger model Llama2-13B. We observed that SUIT attains higher scores than all baselines.

- **Entity-Agnostic Information in $K_s^{\perp}$**: Reviewer U4G2 questioned whether $K_s^{\perp}$ captures entity-agnostic information. To answer this, we conducted an additional controlled experiment where we compared how much the representation in each subspace ($K_s$ and $K_s^{\perp}$) changes when we swap only the entity versus only the context. The results showed that the entity effect is more than twice as large in $K_s$ as in $K_s^{\perp}$, providing direct, quantitative evidence that $K_s$ captures entity-specific variation while $K_s^{\perp}$ largely suppresses it.

- **SUIT not consistently SOTA**: Reviewers noted that SUIT does not consistently achieve SOTA performance (anrX, U4G2). This impression mainly comes from zsRE in Table 1, where SUIT does not substantially outperform AlphaEdit on the Specificity score. This is due to the limitation of the metric; for zsRE, Specificity was measured only by checking whether the first token of the post-edit answer matches the reference. For more rigorous and realistic assessment, we have conducted additional evaluation to measure Specificity by taking the pre-edit model’s answer as the reference and measuring up to 4-token matching. Under these stricter and more goal-aligned metrics, SUIT’s advantage becomes clear: with the 4-token criterion on zsRE, LLaMA3 shows 18.3% preservation with AlphaEdit vs 39.8% with SUIT, and GPT-J shows 65.3% (AlphaEdit) vs 80.3% (SUIT).

---

> ### Author Response · Authors · 2025-12-03
> **Review Summary**
>
> - **$S$ metric and the Generalization–Specificity trade-off (anrX, U4G2)**: In response to the concerns raised by reviewers, we additionally conducted systematic hyperparameter sweeps on both AlphaEdit and SUIT. The results demonstrate that (i) SUIT’s higher $S$ scores are not simply the result of a single inflated Specificity term, and (ii) the Generalization gap observed in Table 1 is not a structural limitation of SUIT, but rather reflects a particular operating point along the Generalization–Specificity trade-off.
>
> - **Clarifying the Lower Baseline Scores (JqDj, anrX)**: Reviewers noted that the baseline scores in Table 1 appear lower than those reported in the original papers. We clarified that this is because we adopt a more rigorous and practical evaluation: (1) for accurate measurement, we only evaluate cases where the pre-edited model actually knows the target fact, and (2) we apply a stricter, generation-based success criterion that better reflects real model usage (with results under the original evaluation setup also reported in the Appendix F.2).
>
> ---
> Lastly, we summarize the key strengths of our work using quotes from the reviewers.
>
> **Principled subspace-editing methodology with clear theoretical grounding**
>
> - “*provides a clear theoretical framing grounded in the Linear Representation Hypothesis, establishing a principled link between semantic subspaces and factual updates. This theoretical foundation not only motivates the method design but also enhances interpretability and conceptual rigor.*” (JqDj)
>
> - “*The paper presents a clean and principled subspace isolation method*” (anrX)
>
> **Strong, empirically well-supported performance**
> - “*These analyses provide both empirical depth and transparency, showing that the improvements are consistent and well-supported.*” (JqDj)
>
> - “*The method demonstrates a significant specificity boost on the COUNTERFACT dataset based on a generation-based criterion.*” (bGWe)
> - “*The experimental results suggest that the proposed method is comparable to SOTA baselines.*” (anrX)
>
> - “*The performance of SUIT is good and surpasses other baselines across datasets.*” (U4G2)
>
> **Mechanistic and empirical support for reduced perturbation**
> - “*This is simple, effective, and empirically validated (variance analysis in Table 2; reduced interaction with the common subspace in Table 3)*” (anrX)
>
> - “*The paper provides mechanistic evidence (e.g., token-level perturbations in residual streams) to validate the reduced perturbation.*” (bGWe)
>
> **Clarity and quality of presentation**
> - “*the paper's presentation is clear and well-organized.*” (U4G2)
>
> Thank you.

---

### Meta-Review · Area_Chair_bnuV · 2026-01-08

**Summary:**

Reviewers generally acknowledged the reasonableness and effectiveness of the proposed method. Concerns regarding experimental issues have been clarified in the author rebuttal. While the method appears somewhat incremental compared to the previous approach, AlphaEdit, it is sufficiently distinct and demonstrates unique advantages—particularly a favorable tradeoff between generalization and specificity—in the experimental results. In my opinion, the paper meets the borderline accept threshold for ICLR.

**Reviewer Concerns:**

It appears that concerns regarding experimental evaluations—including hyperparameters, evaluation settings, inconsistencies with previously reported results, and computational cost—have been well addressed. No significant issues remain outstanding.

**Reviewer Scores:**

All reviewers initially assigned a rating of 4. However, the author rebuttal provides convincing clarifications and additional experimental results that support the effectiveness of the proposed method and its claims. Therefore, I anticipate that some reviewers, particularly U4G2 and anrX, may increase their ratings from 4 to 6, resulting in an overall average score of 5.

---

### Decision · Program_Chairs · 2026-01-26

Accept (Poster)